PREPARED FOR SUBMISSION TO SCIPOST PHYSICS LECTURE NOTES

# Les Houches lectures on non-perturbative topological strings

**Marcos Mariño**

*Département de Physique Théorique et Section de Mathématiques*
*Université de Genève, Genève, CH-1211 Switzerland*

*E-mail:* marcos.marino@unige.ch

ABSTRACT: In these lecture notes for the Les Houches School on Quantum Geometry I give an introductory overview of non-perturbative aspects of topological string theory. After a short summary of the perturbative aspects, I first consider the non-perturbative sectors of the theory as unveiled by the theory of resurgence. I give a self-contained derivation of recent results on non-perturbative amplitudes, and I explain the conjecture relating the resurgent structure of the topological string to BPS invariants. In the second part of the lectures I introduce the topological string/spectral theory (TS/ST) correspondence, which provides a non-perturbative definition of topological string theory on toric Calabi–Yau manifolds in terms of the spectral theory of quantum mirror curves.

## 1 Introduction

This is a written version of the lectures I gave at Les Houches school on Quantum Geometry, in August 2024. My goal in these notes, as in the original lectures, is to provide an introduction to non-perturbative aspects of the topological string. Before embarking on this topic it is useful to have a general view of what is meant by a "non-perturbative" approach, so let me start these notes with a discussion of perturbative *versus* non-perturbative physics in quantum theories.

### 1.1 Perturbative and non-perturbative physics

Observables in physical theories are often functions $F(g)$ of a control parameter $g$ or "coupling constant". There are many situations in which determining $F(g)$ for arbitrary values of $g$ is in practice difficult. If $F(g)$ is known for a reference value of $g$ (which I will take to be $g = 0$), one can try to use perturbation methods to understand what happens when $g$ is near this reference

value, i.e. when $g$ is small. The outcome of these methods is a perturbative series in $g$, of the form

$$\varphi(g) = \sum_{n \geq 0} a_n g^n. \tag{1.1}$$

(In these lectures, $\varphi(g)$ will denote a formal power series.) However, more often than not, the series obtained in perturbation theory are factorially divergent, i.e. the coefficients grow like $a_n \sim n!$. This means that the series $\varphi(g)$ does not define a function in a neighbourhood of $g = 0$. Rather, $\varphi(g)$ provides an asymptotic approximation to $F(g)$, in the sense of Poincaré, and we write

$$F(g) \sim \varphi(g). \tag{1.2}$$

Extracting physical information on $F(g)$ from its asymptotic expansion $\varphi(g)$ has been an important problem in physics and mathematics. A standard technique is to use an optimal truncation of the asymptotic series, but this only gives *approximate* results. In some cases one can obtain exact results by an appropriate resummation of the perturbative series, and by adding non-perturbative effects.

In the above discussion I have assumed that, in our theory, $F(g)$ can be mathematically defined as an actual function, at least for some range of values of $g$. If this is the case, $F(g)$ is called a *non-perturbative definition* of our observable. Usually the perturbative series $\varphi(g)$ can be obtained from the non-perturbative definition. However, as one considers quantum theories of increasing complexity, non-perturbative definitions become harder to obtain. Let us discuss various possible scenarios and their realizations in physical theories.

In the best possible scenario, we have a rigorous mathematical definition of the function $F(g)$, an algorithmic procedure to calculate it for a wide range of values of $g$, and a method to obtain a perturbative expansion for small $g$. This is often the case in quantum mechanics. Let us consider for example a non-relativistic particle in a potential of the form,

$$V(q) = \frac{q^2}{2} + gq^4, \tag{1.3}$$

where the coupling constant $g$ is the strength of the anharmonic, quartic perturbation. A typical observable in this system is e.g. the energy of the ground state $E_0(g)$. When $g > 0$ this function is defined rigorously by the spectral theory of the self-adjoint Schrödinger operator

$$\mathsf{H} = \frac{\mathsf{p}^2}{2} + V(\mathsf{q}), \tag{1.4}$$

where $\mathsf{q}$, $\mathsf{p}$ are canonically conjugate Heisenberg operators. We also have numerical techniques, like Rayleigh–Ritz methods, to compute this ground state energy. Finally, the rules of stationary perturbation theory give a power series in $g$ for $E_0(g)$, of the form

$$\varphi(g) = \frac{1}{2} + \frac{3g}{4} - \frac{21}{8}g^2 + \cdots \tag{1.5}$$

It can be shown that this gives indeed an asymptotic expansion for $E_0(g)$. Moreover, one can use Borel resummation techniques to recover the exact $E_0(g)$ from $\varphi(g)$ (see e.g. [1] for a review and references).

In the second best scenario, one has a method to obtain perturbative expansions, and an non-rigorous algorithmic procedure to calculate $F(g)$ non-perturbatively. This is the typical situation in quantum field theory (QFT). One of the main achievements in QFT is the development

of renormalized perturbation theory, which produces mathematically well-defined formal power series in a coupling constant $g$. For some observables we can also obtain a non-perturbative definition by using a lattice regularization of the path integral, and then taking the continuum limit. This latter procedure is in general not mathematically rigorous, but in practice seems to leads to well-defined and explicit numerical results. There is a branch of mathematical physics, called constructive QFT, whose goal is to provide mathematically rigorous non-perturbative definitions of observables, akin to what can be achieved in quantum mechanics. Recently there have been some advances in constructive QFT by using probabilistic techniques, but progress in that front has been slow and mostly in low dimensions. There are special cases in QFT in which we can obtain non-perturbative approaches by other means. For example, in integrable quantum field theories one can use the Bethe ansatz and form factor expansions to obtain non-perturbative definitions of some observables. In some theories with a large $N$ expansion one can sometimes obtain exact results as a function of the renormalized coupling constant, albeit order by order in a series in $1/N$. An additional difficulty of QFTs is that the relationship between the perturbative and the non-perturbative approaches is more complicated than in quantum mechanics, and showing that the perturbative series provides an asymptotic expansion of the available non-perturbative definitions becomes non-trivial.

The case of string theories is even more challenging, since they are defined only by perturbative expansions, and non-perturbative definitions simply do not exist in general. One can try to construct string field theories, i.e. spacetime actions whose Feynman rules reproduce ordinary perturbation theory. There are simpler examples of string theories in low dimensions where one can use a sort of lattice regularization in terms of matrix integrals in order to define exact observables. Another class of examples concerns superstring theories on Anti-de Sitter (AdS) backgrounds, which are expected to be dual to a superconformal QFT. In these backgrounds, the non-perturbative definition of string theory amounts to the non-perturbative definition of the dual QFT.

Therefore, both in QFT and in string theory we have in principle a systematic approach to compute formal power series in the coupling constant, through the rules of perturbation theory. We have a harder time in obtaining non-perturbative, exact definitions of observables. This problem becomes particularly acute in string theory. In view of this, it might be a good idea to try to extract as much information as possible from the perturbative series itself, as 't Hooft advocated [2]. It turns out that there is a framework to do this which was started in the late 1970s-early 1980s by various physicists, and it was later formalized by the mathematician Jean Écalle under the name of theory of resurgence. The basic idea of the theory of resurgence is that one can obtain non-perturbative results by appropriate resummations of formal series. However, in order to do that one needs to go beyond the perturbative sector and to consider *non-perturbative effects*, which mathematically are formal power series with an additional exponentially small dependence on the coupling constant. Some of these non-perturbative effects (but in general not all) turn out to be hidden in the perturbative series, and one can learn something about the non-perturbative aspects of the theory by extracting these effects from perturbation theory.

In these lectures, our starting point will be a perturbative series $\varphi(g)$, and the search for "non-perturbative" aspects will refer to either of the following two problems:

1. *Non-perturbative effects*: given $\varphi(g)$, can we obtain an explicit description of the non-perturbative effects which are hidden in the perturbative series?

2. *Non-perturbative definition*: given $\varphi(g)$, is it possible to construct a well-defined function $F(g)$ which has $\varphi(g)$ as its asymptotic expansion?

These two problems are logically independent and they have a different flavour. The first problem has a unique solution. It is encoded in the so-called *resurgent structure* associated to the perturbative series $\varphi(g)$ (resurgent structures were introduced in [3] and their definition is presented in the Appendix A). However, obtaining explicit descriptions of resurgent structures turns out to be a very difficult problem, even in simple quantum theories. The second problem clearly does not have a unique solution, since there are infinitely many functions with the same asymptotic expansion. Therefore, the relevant question is whether there is a way to pick up a particular non-perturbative definition as the one which is physically relevant, or perhaps the one which is more interesting mathematically.

Although the two problems above can be addressed separately, they are also related, in the sense that once a solution to the second problem has been found and a non-perturbative definition is available, one can ask whether it can be reconstructed by using the resurgent structure, i.e. by using the non-perturbative effects obtained in the solution to the first problem.

## 1.2   Non-perturbative topological strings

Topological string theory can be regarded as a simplified model of string theory, more complex than non-critical string theories, but still simpler than full-fledged string theories. Topological string theories are interesting for various reasons. They provide a physical counterpart to the theory of enumerative invariants for Calabi–Yau (CY) threefolds, and they have led to many surprising results in that field. Through the idea of geometric engineering [4], they are closely related to $\mathcal{N} = 2$ supersymmetric gauge theories in four and five dimensions. They have multiple connections to classical and quantum integrable models, as seen in e.g. [5–8]. Finally, they lead to simpler but precise realizations of large $N$ dualities, in which the large $N$ dual can be a matrix model [9–12], a Chern–Simons gauge theory [13], or, as we will see in these lectures, a one-dimensional quantum-mechanical model [14, 15]. For these reasons, there have been various efforts to understand non-perturbative aspects of topological strings from different perspectives. In these lectures I will address this problem by first considering the resurgent structure of topological strings, as proposed in question 1 above, and then addressing the question 2 of finding an interesting non-perturbative definition.

I have tried to preserve three aspects of the original lectures. First, there are many references to the other lectures of the school, and the interested reader can consult the webpage https://houches24.github.io for further information. Second, I have included many exercises which complement the exposition. *Mathematica* programs with solutions to some of the exercises can be found in my webpage https://www.marcosmarino.net/lecture-notes.html. Finally, although the lectures are detailed, they are also informal and pedagogical, and the focus is often on detailed case studies, rather than on general constructions.

The structure of these lectures is the following. I will first review some properties of perturbative topological strings in section 2. In section 3 I address their resurgent structure, and I will essentially answer question 1 above in the case of topological string theory. In section 4, I provide a possible answer to question 2 in the case of toric CY threefolds, namely, I consider a well-defined function, obtained from a quantum-mechanical problem, whose asymptotic expansion conjecturally reproduces the perturbative series of the topological string. Appendix A is a quick summary of the theory of resurgence. Appendix B lists some properties of Faddeev's quantum dilogarithm which are used in the lectures.

## 2 Perturbative topological strings

In this section I give some background on perturbative topological strings. This is a big subject which would require many lectures in itself, so I cannot cover all the details. Useful references on topological string theory and mirror symmetry include [16–21]. Various aspects of the mathematical approach to topological strings have been presented in the lectures of M. Liu in this school. What I do in this section is essentially to provide a list of properties of topological string theory which will be useful later on.

Topological string theory, as I mentioned above, can be regarded as a toy model of string theory. Since a string theory is a quantum theory of maps from Riemann surfaces to a target manifold, we have so specify first our target. This will be a CY threefold, i.e. a complex, Kähler, Ricci-flat manifold of complex dimension three (other choices are possible, but have been studied less intensively). I will denote such a target by $M$, and I would like to emphasize that $M$ does not have to be compact. In fact, non-compact CY threefolds will be very important for us, for various reasons.

The starting point to construct topological string theory is the $\mathcal{N} = 2$ supersymmetric version of the non-linear sigma model, with target space $M$. This model can be topologically twisted to obtain a topological field theory in two dimensions [22]. Topological string theory is then obtained by coupling the resulting twisted theory to topological gravity, in the way explained in [23, 24]. There are however two different ways of twisting the non-linear sigma model, known as the A and the B twist [25, 26], and this leads to two different versions of topological string theory, which are usually called the A and the B model. These models are sensitive to different properties of $M$. The A model is sensitive to the Kähler parameters of $M$, which specify the (complexified) sizes of the two-cycles in $M$. There are $h^{1,1}(M) = b_2(M)$ Kähler parameters in total, where $h^{p,q}(M)$ are the Hodge numbers of $M$, and $b_i(M)$ are its Betti numbes. We will denote these parameters by $t_i$, $i = 1, \cdots, s$, where $s = h^{1,1}(M)$, and we will gather them in a vector $\mathbf{t} = (t_1, \cdots, t_s)$. The B model is sensitive to the complex parameters of $M$, which specify its "shape", and there are

$$h^{1,2}(M) = \frac{b_3(M)}{2} - 1 \tag{2.1}$$

complex moduli in total. We will denote them by $z_i$, $i = 1, \cdots, h^{1,2}(M)$.

*Mirror symmetry* is a duality or equivalence between the A model on the CY manifold $M$ and the B model on the mirror CY $M^\star$. This means in particular that there is a map between the Kähler and the complex moduli of the mirror manifolds, which we can write as $t_i = t_i(z_1, \cdots, z_s)$, $i = 1, \cdots, s$. This map is usually called the *mirror map*, and we will see explicit examples below.

Topological string theory and mirror symmetry were formulated originally for compact CY manifolds. However, there is a class of CY manifolds, called toric manifolds, which admit a torus action and are non-compact. An example of such a toric CY is given by the total space of the canonical bundle of $\mathbb{P}^2$,

$$X = \mathcal{O}(-3) \to \mathbb{P}^2, \tag{2.2}$$

which appeared in the lectures by V. Bouchard [27] and also by M. Liu. Other examples of toric manifolds are obtained by considering the total space of the canonical bundle of appropriate algebraic surfaces, like $\mathbb{P}^1 \times \mathbb{P}^1$. In spite of their non-compactness, it is possible to define the A model on toric CY manifolds [4, 28, 29]. In particular, there is a version of mirror symmetry for toric CY manifolds called *local mirror symmetry* which is particularly simple and useful. We will discuss some aspects of local mirror symmetry in more detail below.

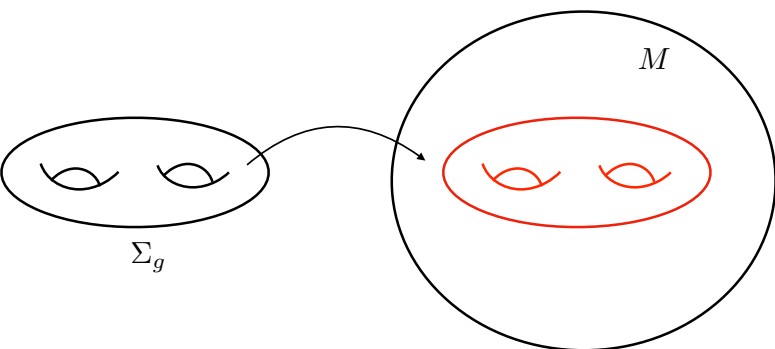

**Figure 1**: A pictorial representation of a holomorphic map from a Riemann surface $\Sigma_g$ into a CY $M$.

The only observable of topological string theory on a CY threefold is the partition function, or its logarithm the free energy. The latter can be calculated as a perturbative series by summing over connected Riemann surfaces. The contribution of a genus $g$ Riemann surfaces to the free energy will be denoted by $F_g$, and it is a function of the Kähler (respectively, complex) moduli in the A (respectively, B) model.

### 2.1 The A model

To solve topological string theory perturbatively, one has to calculate $F_g$ for all $g \geq 0$. Let us describe how to do this calculation in the A model. Since the theory is topological, one can show that it just "counts" instantons of the twisted non-linear sigma model with target $M$. The instantons are in this case holomorphic maps from the Riemann surface of genus $g$ to the CY $M$,

$$f : \Sigma_g \to M, \tag{2.3}$$

see Fig. 1 for a pictorial representation. Let $[S_i] \in H_2(M, \mathbb{Z})$, $i = 1, \cdots, s$, be a basis for the two-homology of $M$, with $s = b_2(M)$ as before. The maps (2.3) are classified topologically by the homology class

$$f_*[(\Sigma_g)] = \sum_{i=1}^{s} d_i[S_i] \in H_2(X, \mathbb{Z}), \tag{2.4}$$

where $d_i$ are integers called the *degrees* of the map. We will put them together in a degree vector $\mathbf{d} = (d_1, \cdots, d_s)$. The "counting" of instantons is given by the *Gromov–Witten (GW) invariant* at genus $g$ and degree $\mathbf{d}$, which we will denote by $N_g^{\mathbf{d}}$, and is given by an appropriate integral over the space of collective coordinates of the instanton, or moduli space of maps, as explained in M. Liu's lectures (see also [18] for definitions and examples). Note that GW invariants are in general rational, rather than integer, numbers.

The genus $g$ free energies $F_g(\mathbf{t})$ can be computed as an expansion near the so-called *large radius point* $\mathbf{t} \to \infty$, involving the GW invariants at genus $g$ and at all degrees. They are given by formal power series in $\mathrm{e}^{-t_i}$, $i = 1, \cdots, s$, where $t_i$ are the Kähler parameters of $M$. They also involve additional contributions, which are polynomials in the $t_i$. At genus zero, the free energy reads

$$F_0(\mathbf{t}) = \frac{1}{6} \sum_{i,j,k=1}^{s} a_{ijk} t_i t_j t_k + \sum_{\mathbf{d}} N_{0,\mathbf{d}} \mathrm{e}^{-\mathbf{d} \cdot \mathbf{t}}. \tag{2.5}$$

In the case of a compact CY threefold, the numbers $a_{ijk}$ are interpreted as triple intersection numbers of two-classes in $X$, and one usually adds an additional polynomial of degree two in the Kähler parameters, but we will not busy ourselves with these details here. At genus one, one has

$$F_1(\mathbf{t}) = \sum_{i=1}^{s} b_i t_i + \sum_{\mathbf{d}} N_{1,\mathbf{d}} e^{-\mathbf{d}\cdot\mathbf{t}}. \tag{2.6}$$

In the compact case, the coefficients $b_i$ are related to the second Chern class of the CY manifold [30]. At higher genus one finds

$$F_g(\mathbf{t}) = c_g \chi + \sum_{\mathbf{d}} N_{g,\mathbf{d}} e^{-\mathbf{d}\cdot\mathbf{t}}, \qquad g \geq 2. \tag{2.7}$$

Here, $\chi$ is the Euler characteristic of $M$ in the compact case, and a suitable generalization thereof in the non-compact case. The constant term $c_g \chi$ in (2.7) is the contribution of *constant maps* to the genus $g$ free energy. The coefficient $c_g$ is given by an integral over the moduli space of Riemann surfaces [24], whose value can be obtained by using string dualities [31] or by a direct calculation [32],

$$c_g = \frac{(-1)^{g-1} B_{2g} B_{2g-2}}{4g(2g-2)(2g-2)!}. \tag{2.8}$$

Let us mention that the infinite sums defining the free energies are sums over instantons of the underlying topological sigma model. The exponent $\mathbf{d} \cdot \mathbf{t}/\ell_s^2$ is the action of an instanton whose topological class is labelled by the degrees $\mathbf{d}$, and we have included the string length $\ell_s$, which is usually set to one.

Although the genus $g$ free energies have been written in (2.5), (2.6), (2.7) as formal power series, they have a common region of convergence near the large radius point $t_i \to \infty$, and therefore they define actual functions $F_g(\mathbf{t})$, at least near that point in moduli space. The total free energy of the topological string is formally defined as the sum,

$$F(\mathbf{t}; g_s) = \sum_{g \geq 0} g_s^{2g-2} F_g(\mathbf{t}). \tag{2.9}$$

This can be further decomposed as

$$F(\mathbf{t}; g_s) = F^{(\mathrm{p})}(\mathbf{t}; g_s) + \sum_{g \geq 0} \sum_{\mathbf{d}} N_{g,\mathbf{d}} e^{-\mathbf{d}\cdot\mathbf{t}} g_s^{2g-2}, \tag{2.10}$$

where

$$F^{(\mathrm{p})}(\mathbf{t}; g_s) = \frac{1}{6g_s^2} \sum_{i,j,k=1}^{s} a_{ijk} t_i t_j t_k + \sum_{i=1}^{s} b_i t_i + \chi \sum_{g \geq 2} c_g g_s^{2g-2}. \tag{2.11}$$

is the polynomial part of the free energies. The variable $g_s$, called the *topological string coupling constant*, is in principle a formal variable, keeping track of the genus of the Riemann surface. However, in string theory this constant has a physical meaning, and measures the strength of the string interaction: when $g_s$ is very small, the contribution to the free energy is dominated by Riemann surfaces of low genus; as $g_s$ becomes large, the contribution of higher genus Riemann surfaces becomes important.

If we fix the value of $\mathbf{t}$ inside the common radius of convergence of the free energies, the sum over genera in (2.9) defines a formal power series in the string coupling constant, whose

coefficients $F_g(\mathbf{t})$ are functions of the moduli. Understanding the detailed properties of this series will be one of the central goals of these lectures. There is strong evidence that the series $F_g(\mathbf{t})$, at a fixed value of $\mathbf{t}$, diverges doubly-factorially,

$$F_g(\mathbf{t}) \sim (2g)!, \tag{2.12}$$

therefore the total free energy (2.9) does *not* define a function of $g_s$ and $\mathbf{t}$. The factorial divergence of the genus expansion is typical of perturbative series in quantum theories, as we mentioned in the introduction. In the case of string theory, it was first noted for the bosonic string in [33]. Shenker argued in [34] that the behavior (2.12) should be a generic feature of any string theory, due to the growth of the "Feynman integrals" on the moduli space of Riemann surfaces of genus $g$ that compute the relevant string theory amplitudes. Very recently, the double-factorial growth was proved in [35] for the free energies obtained from topological recursion.

**Remark 2.1.** In quantum field theory one typically distinguishes between two different sources for the factorial growth of perturbation theory. The first source is due to the growth of Feynman diagrams and is related to instantons. The second source is due to the integration over momenta in some special diagrams, and the corresponding Borel singularities are called "renormalons." In the case of string theory this distinction becomes more subtle. Since there is only one diagram at each genus, we could say that the factorial growth of string theory is due to integration over moduli, and therefore is of the renormalon type. At the same time, as pointed out in [34], one can use Feynman diagrams to study these integrals, and relate them in many cases to instanton configurations in matrix models.

## 2.2 The Gopakumar–Vafa representation

The expression in the r.h.s. of (2.10) is a double expansion, in both degrees and genera. To define the total free energy (2.9) we first sum over all the degrees at fixed genus, to obtain the functions $F_g(\mathbf{t})$, and then we sum over genera. As we will see, this is the natural answer that one obtains from mirror symmetry and the B-model. But perhaps one could try to exchange the order of summations, i.e. to sum over all genera for a fixed $\mathbf{d}$, and them sum over all degrees. This representation can be obtained by using the results of Gopakumar and Vafa in [36], who reformulated the total free energy of topological string theory by using a physical realization in type IIA superstring theory and M-theory. Let us consider the double expansion in (2.10) involving the GW invariants,

$$\sum_{g \geq 0} \sum_{\mathbf{d}} N_{g,\mathbf{d}}\, e^{-\mathbf{d} \cdot \mathbf{t}}\, g_s^{2g-2}. \tag{2.13}$$

Then, [36] found that this series can be re-expressed as

$$F^{\mathrm{GV}}(\mathbf{t}; g_s) = \sum_{g \geq 0} \sum_{\mathbf{d}} \sum_{w=1}^{\infty} \frac{1}{w} n_g^{\mathbf{d}} \left(2 \sin \frac{w g_s}{2}\right)^{2g-2} e^{-w \mathbf{d} \cdot \mathbf{t}}, \tag{2.14}$$

where $n_g^{\mathbf{d}}$ are the so-called *Gopakumar–Vafa (GV) invariants*. In contrast to the GW invariants, they turn out to be *integer* numbers. They can be interpreted, roughly, as Euler characteristics of moduli spaces of D2 branes in the CY manifold (see e.g. [37] for a detailed discussion with examples). One important property of the GV invariants is that, for a given degree $\mathbf{d}$, there is a maximal genus $g_{\max}(\mathbf{d})$ such that $n_g^{\mathbf{d}} = 0$ for $g > g_{\max}(\mathbf{d})$.

**Exercise 2.2.** Show that the expression (2.14) leads to the following formula for $F_g(\mathbf{t})$ [37]. For $g = 0$, $g = 1$, one has

$$F_0(\mathbf{t}) = F_0^{(\mathrm{p})}(\mathbf{t}) + \sum_{\mathbf{d}} n_0^{\mathbf{d}} \, \mathrm{Li}_3 \left( \mathrm{e}^{-\mathbf{d} \cdot \mathbf{t}} \right),$$

$$F_1(\mathbf{t}) = F_1^{(\mathrm{p})}(\mathbf{t}) + \sum_{\mathbf{d}} \left( \frac{n_0^{\mathbf{d}}}{12} + n_1^{\mathbf{d}} \right) \mathrm{Li}_1 \left( \mathrm{e}^{-\mathbf{d} \cdot \mathbf{t}} \right),$$

(2.15)

while for $g \geq 2$ one has

$$F_g(\mathbf{t}) = F_g^{(\mathrm{p})}(\mathbf{t})$$
$$+ \sum_{\mathbf{d}} \left( \frac{(-1)^{g-1} B_{2g} n_0^{\mathbf{d}}}{2g(2g-2)!} + \frac{2(-1)^g n_2^{\mathbf{d}}}{(2g-2)!} + \cdots - \frac{g-2}{12} n_{g-1}^{\mathbf{d}} + n_g^{\mathbf{d}} \right) \mathrm{Li}_{3-2g} \left( \mathrm{e}^{-\mathbf{d} \cdot \mathbf{t}} \right).$$

(2.16)

In these expressions,

$$\mathrm{Li}_n(z) = \sum_{k \geq 1} \frac{z^k}{k^n}$$

(2.17)

is the polylogarithm function of order $n$. $\qquad \qquad \square$

It follows from (2.16) that if one knows the GW invariants $N_{g',\mathbf{d}'}$ with $g' \leq g$, $\mathbf{d}' \leq \mathbf{d}$, one can determine uniquely the GV invariant $n_g^{\mathbf{d}}$, and viceversa. In that sense, the two sets of invariants contain the same information. Let us note that there exist direct mathematical constructions of the GV invariants as well, see e.g. [38].

The GV representation of the total free energy gives another view on the problem of resummation. It is clear that we can now write the non-trivial part of the total free energy, involving the GW invariants, as

$$F^{\mathrm{GV}}(\mathbf{t}; g_s) = \sum_{\mathbf{m}} F_{\mathbf{m}}(g_s) \mathrm{e}^{-\mathbf{m} \cdot \mathbf{t}},$$

(2.18)

where

$$F_{\mathbf{m}}(g_s) = \sum_{g \geq 0} \sum_{\mathbf{m} = w\mathbf{d}} \frac{1}{w} n_g^{\mathbf{d}} \left( 2 \sin \frac{w g_s}{2} \right)^{2g-2}.$$

(2.19)

Note that, due to the vanishing property of the GV invariants mentioned above, the sum over $g$ in (2.19) is finite. One could think that (2.18) can perhaps be summed, as a series now in the variables $\mathrm{e}^{-t_i}$. It turns out that the properties of this series depend crucially on the value of $g_s$. If $g_s$ is *real*, the series is not even well-defined. This is due to the inverse square sines appearing in (2.14), which lead to singularities at rational values of $g_s$. In fact, given any rational value of $g_s$, there is a minimum degree $\mathbf{m}_{\min}$ such that infinitely many coefficients $F_{\mathbf{m}}(g_s)$ with $\mathbf{m} > \mathbf{m}_{\min}$ are singular at that rational value. As a consequence, given any real value of $g_s$, rational or not, there is a degree starting from which infinitely many coefficients $F_{\mathbf{m}}(g_s)$ can be made arbitrarily large. This is clearly a pathological situation. One could still try to make sense of the r.h.s. of (2.18) for complex values of $g_s$. However, in that case there is evidence that the coefficients $F_{\mathbf{m}}(g_s)$ grow with Gargantuan speed, even worst than factorial. For example, in the case of local $\mathbb{P}^2$, introduced in (2.2), one has $b_2(M) = 1$ (corresponding to the $\mathbb{P}^1$ inside $\mathbb{P}^2$) and therefore there is a single degree. Explicit computations suggest that [39, 40]

$$\log |F_m(g_s)| \sim m^2, \qquad m \gg 1,$$

(2.20)

at least when $|g_s|$ is not too small, $|g_s| \gtrsim 1$. This growth seems to be the consequence of two facts. First, for complex $g_s$, the sin factors in the GV formula become hyperbolic functions and they lead to contributions in $F_m(g_s)$ of the form $c^{2g}$, with $|c| > 1$ for $|g_s| \gtrsim 1$. Second, the maximal genus of a curve of degree $m$ in a CY grows like $g(m) \sim m^2$, as a consequence of Castelnuovo theory [41], which is precisely the behavior in (2.20).

Therefore, when $g_s$ is real the series (2.18) does not make sense, and when $g_s$ is complex it makes sense, but it diverges wildly. This underappreciated fact shows that in general the GV representation of the free energy does not define topological string theory non-perturbatively.

**Exercise 2.3.** M. Liu's lectures have introduced the topological vertex of [42], which can be used to compute the coefficients $F_m(g_s)$ efficiently in the case of local $\mathbb{P}^2$. One finds, for the very first values of $m = 1, 2, 3$,

$$F_1(g_s) = -\frac{3q^2}{(q^2 - 1)^2},$$

$$F_2(g_s) = \frac{3q^2\left(4q^4 + 7q^2 + 4\right)}{2\left(q^4 - 1\right)^2}, \tag{2.21}$$

$$F_3(g_s) = -\frac{10q^{12} + 27q^{10} + 54q^8 + 62q^6 + 54q^4 + 27q^2 + 10}{\left(q^6 - 1\right)^2},$$

where $q = \mathrm{e}^{\mathrm{i}g_s/2}$. Write a computer program which calculates these coefficients, and extract from them the very first GV invariants of this geometry. Verify the asymptotic behaviour (2.20). $\square$

**Remark 2.4.** There is a special class of toric CY manifolds which can be used to engineer five-dimensional $SU(N)$ gauge theories [4, 43]. For these manifolds, the gauge theory instanton partition function (reviewed in N. Nekrasov's lectures in this school) provides a rearrangement of the GV expansion which converges for complex $g_s$ [44]. However, it is still singular if $g_s \in 2\pi\mathbb{Q}$.

## 2.3   An example: the resolved conifold

Before going on, let us consider what is perhaps the simplest example of a topological string theory, on the non-compact CY manifold known as the *resolved conifold*. This manifold is a plane bundle over the two-sphere:

$$X = \mathcal{O}(-1) \oplus \mathcal{O}(-1) \to \mathbb{P}^1. \tag{2.22}$$

There is a single modulus $t$, which in the A-model is the (complexified) area of the $\mathbb{P}^1$. A calculation of the Gromov–Witten invariants in [32] shows that there is a single non-zero GV invariant in this geometry, with $g = 0$ and $d = 1$, and equal to $n_0^1 = 1$. The all-genus free energy of the topological string, in the GV representation, is then simply given by

$$F^{\mathrm{GV}}(t; g_s) = \sum_{w=1}^{\infty} \frac{1}{w} \frac{\mathrm{e}^{-wt}}{4\sin^2\left(\frac{wg_s}{2}\right)}. \tag{2.23}$$

The full free energy differs from this expression in a cubic polynomial in $t$ which is not important for the discussion. Up to such a polynomial, one finds for the free energies at fixed genus,

$$F_0(t) = \mathrm{Li}_3(\mathrm{e}^{-t}),$$

$$F_1(t) = \frac{1}{12}\mathrm{Li}_1(\mathrm{e}^{-t}), \tag{2.24}$$

$$F_g(t) = \frac{(-1)^{g-1}B_{2g}}{2g(2g-2)!}\mathrm{Li}_{3-2g}(\mathrm{e}^{-t}), \qquad g \geq 2.$$

These can be obtained from the general expression (2.16) by taking into account that all GV invariants vanish except $n_0^1 = 1$. It is easy to see that the sequence $F_g(t)$ grows doubly-factorially with the genus, by using the formula

$$\mathrm{Li}_{3-2g}(\mathrm{e}^{-t}) = \Gamma(2g-2) \sum_{k\in\mathbb{Z}} \frac{1}{(2\pi k\mathrm{i} + t)^{2g-2}}, \tag{2.25}$$

which is valid for $g \geq 2$ and $\mathrm{e}^{-t} \neq 1$.

An even simpler topological string theory can be obtained when we look at the limit of $F_g(t)$ as $t \to 0$ and we keep the most singular terms. One finds,

$$
\begin{aligned}
F_0(\lambda) &= \frac{\lambda^2}{2}\left(\log(\lambda) - \frac{3}{2}\right) + \mathcal{O}(1), \\
F_1(\lambda) &= -\frac{1}{12}\log(\lambda) + \mathcal{O}(1), \\
F_g(\lambda) &= \frac{B_{2g}}{2g(2g-2)}\lambda^{2-2g} + \mathcal{O}(1), \quad g \geq 2.
\end{aligned}
\tag{2.26}
$$

where $\lambda = \mathrm{i}t$. The point $t = 0$ (or $\lambda = 0$) is a special point in the moduli space of the resolved conifold, in which the area of the $\mathbb{P}^1$ shrinks to zero size, and the free energy is singular. Such a point in moduli space is called a *conifold point*. Conifold points arise generically in the moduli space of CY manifolds, and they will play an important role in what follows.

## 2.4 The B model

The problem of calculating the free energies in the B model is very different, since the twisted sigma model localizes to constant maps [25], so the calculation is in a sense "classical". In the case of genus zero, the problem is completely solved by calculating the periods of the holomorphic 3-form $\Omega$ on the CY $M^\star$, as explained in the pioneering paper [45]. One chooses a symplectic basis of three-cycles,

$$A^I, \ B_I, \qquad I = 0, 1, \cdots, s, \tag{2.27}$$

which satisfy

$$
\begin{aligned}
\langle A^I, A^J \rangle &= \langle B^I, B^J \rangle = 0, \\
\langle A^I, B_J \rangle &= -\langle B_I, A^J \rangle = \delta^I_J,
\end{aligned}
\tag{2.28}
$$

where $I, J = 0, 1, \cdots, s$ and $\langle \cdot, \cdot \rangle$ is the intersection pairing in $H_3(M^\star)$. Integration of $\Omega$ over these cycles gives the A and B periods,

$$X^I = \int_{A^I} \Omega, \qquad \mathcal{F}_I = \int_{B_I} \Omega. \tag{2.29}$$

The periods are used to define a projective prepotential $\mathfrak{F}_0(X^I)$ by the relations

$$\mathcal{F}_I = \frac{\partial \mathfrak{F}_0}{\partial X^I}. \tag{2.30}$$

We can now construct the so-called *flat coordinates* $t_a$ as affine coordinates corresponding to the projective coordinates $X^I$: we choose a nonzero period, say $X^0$, and we consider the quotients

$$t_a = \frac{X^a}{X^0}, \qquad a = 1, \cdots, s. \tag{2.31}$$

Since the projective prepotential is homogeneous, we can define a quantity $F_0(\mathbf{t})$ (called the *prepotential*) which only depends on the coordinates $t_a$:

$$\mathfrak{F}_0(X^I) = (X^0)^2 F_0(\mathbf{t}). \tag{2.32}$$

The prepotential gives the genus zero free energy of the topological string, in the B model. An important bonus of using the B model is that the genus zero free energy is obtained as a *global function* on the moduli space of the CY manifold.

According to mirror symmetry, the B model on $M^\star$ is equivalent to the A model on the mirror manifold $M$. In particular, there is an appropriate choice of the symplectic basis of three-cycles such that the corresponding flat coordinates give the mirror map, i.e. the $t_a$ can be regarded as complexified Kähler coordinates on $M$. For that choice, the prepotential defined above agrees with the genus zero free energy of the A model on $M$. In particular, the expansion of that prepotential around the large radius point leads to the GW invariants of the mirror CY. This is the classical setting of mirror symmetry as first discovered in [45].

As we mentioned above, in the case of toric CY manifolds we have a simpler setting for mirror symmetry, called local mirror symmetry. The equation for the mirror of a toric CY turns out to be of the form

$$uv = P(\mathrm{e}^x, \mathrm{e}^y), \tag{2.33}$$

where $P(\mathrm{e}^x, \mathrm{e}^y)$ is a polynomial in the exponentiated variables $x$, $y$. There is a precise algorithm to obtain this polynomial, starting with the description of a toric CY manifold as a symplectic quotient, see e.g [28, 46]. The geometry of the threefold (2.33) is encoded in the Riemann surface $\Sigma$ described by $P$,

$$P(\mathrm{e}^x, \mathrm{e}^y) = 0. \tag{2.34}$$

It can be shown that, for the CY (2.33), the periods of $\Omega$ reduce to the periods of the differential

$$\lambda = y(x)\mathrm{d}x \tag{2.35}$$

on the curve (2.34) [4, 28]. In addition, one can set $X^0 = 1$. The flat coordinates $\mathbf{t}$ and the genus zero free energy $F_0(\mathbf{t})$ in the large radius frame are determined by choosing an appropriate symplectic basis of one-cycles on the curve, $\mathcal{A}_a$, $\mathcal{B}_a$, $a = 1, \cdots, g_\Sigma$, and one finds

$$t_a = \oint_{\mathcal{A}_a} \lambda, \qquad \frac{\partial F_0}{\partial t_a} = \oint_{\mathcal{B}_a} \lambda, \qquad i = 1, \cdots, g_\Sigma, \tag{2.36}$$

where $g_\Sigma$ is the genus of the mirror curve. We note that, in general, $s \geq g_\Sigma$, and there are additional $s - g_\Sigma$ moduli of the CY that are obtained by considering in addition residues of poles at infinity (these additional parameters are sometimes called *mass parameters*).

Another important consequence of using the B model is that there is in fact an infinite family of flat coordinates and genus zero free energies, depending on the choice of a basis of three cycles. Different choices are related by symplectic transformations. This structure is present in the general, compact case, but in order to make things simpler, I will focus on the local case and in addition I will assume that $g_\Sigma = 1$. Then, a symplectic transformation of the cycles induces the following transformation of the periods,

$$\begin{pmatrix} \partial_t F_0 \\ t \end{pmatrix} \to \begin{pmatrix} \partial_{\tilde{t}} \widetilde{F}_0 \\ \tilde{t} \end{pmatrix} = \begin{pmatrix} \alpha & \beta \\ \gamma & \delta \end{pmatrix} \begin{pmatrix} \partial_t F_0 \\ t \end{pmatrix} + \begin{pmatrix} b \\ a \end{pmatrix}, \tag{2.37}$$

where

$$\alpha\delta - \beta\gamma = 1. \tag{2.38}$$

This transformation is a combination of an $SL(2, \mathbb{R})$ transformation and a shift. The shift is due to the fact that in the local case there is a constant period, independent of the moduli, which can mix with the non-trivial periods. The genus zero free energy transforms as,

$$\widetilde{F}_0(\tilde{t}) = F_0(t) - S(t, \tilde{t}), \tag{2.39}$$

which is a generalized Legendre transform. The function $S(t, \tilde{t})$ has the form

$$S(t, \tilde{t}) = \lambda t^2 + \mu t\tilde{t} + \nu\tilde{t}^2 + \hat{a}t + \hat{b}\tilde{t}. \tag{2.40}$$

The coefficients appearing in this polynomial can be related to the parameters appearing in (2.37) by imposing that $\widetilde{F}_0(\tilde{t})$ is independent of $t$,

$$\frac{\partial F_0}{\partial t} = \frac{\partial S}{\partial t} = 2\lambda t + \mu\tilde{t} + \hat{a}. \tag{2.41}$$

and by using that

$$\frac{\partial \widetilde{F}_0}{\partial \tilde{t}} = -\frac{\partial S}{\partial \tilde{t}}. \tag{2.42}$$

By comparing these two equations to the equations for $\tilde{t}$ and $\partial_{\tilde{t}}\widetilde{F}_0$ from (2.37), one eventually obtains

$$S(t, \tilde{t}) = -\frac{\delta}{2\gamma}t^2 + \frac{1}{\gamma}t\tilde{t} - \frac{\alpha}{2\gamma}\tilde{t}^2 - \frac{a}{\gamma}t + \left(\frac{\alpha}{\gamma}a - b\right)\tilde{t}. \tag{2.43}$$

In this derivation I have assumed that $\gamma \neq 0$. The different choices of genus zero free energy (or of flat coordinate $\tilde{t}$ in (2.36)) are usually called choices of *frame*. They are all related by this type of transformations, and they contain the same information.

The local case has additional advantages due to the "remodeling" or BKMP conjecture [10, 47] mentioned in the lectures by V. Bouchard [27]. According to this conjecture (now a theorem), the higher genus free energies can be obtained through the topological recursion of Eynard–Orantin [48], applied to the spectral curve (2.34), endowed with the differential (2.35). As a consequence, it can be shown that, under a symplectic transformation, the total free energy changes by a generalized Fourier transform [49] (this was first postulated in [50]). Let us write down the result in the simple case with $g_\Sigma = 1$. One has

$$\exp(\widetilde{F}(\tilde{t}; g_s)) = \int \exp\left(F(t; g_s) - \frac{1}{g_s^2}S(t, \tilde{t})\right) dt, \tag{2.44}$$

where the function $S(t, \tilde{t})$ implementing the transform is given by (2.43). The integral in the r.h.s. of (2.44) has to be understood formally, since the total free energy appearing in the integrand is itself a formal power series. To obtain the transformation properties of the genus $g$ free energies, we evaluate the integral in the r.h.s. of (2.44) in a saddle point approximation for $g_s$ small. At leading order we recover the generalized Legendre transform (2.39), and in particular the condition for a saddle point is precisely (2.41). The evaluation at higher orders leads to explicit transformation properties for the higher genus free energies, see [50] for examples.

As we mentioned above, in the moduli space of CY manifolds there are generically conifold loci, which are characterized by the shrinking of a three-cycle with the topology of a three-sphere $\mathbb{S}^3$, and lead to a vanishing period (in the local case we have a vanishing one-cycle in the

mirror curve). Like before, we will restrict for simplicity to CYs with a one-dimensional moduli space, where the conifold locus is just a conifold point. There is a particularly important frame in topological string theory defined by the property that the local coordinate is the vanishing period at the conifold point (up to an overall normalization). This frame is called the *conifold frame*. It turns out that the genus $g$ free energies in that frame have the universal behaviour (2.26) near the conifold point, for an appropriate normalization of the vanishing period $\lambda$. This is an important property of topological strings first noted in [51], where a physical explanation of this behaviour was also proposed. In the case of the resolved conifold, the conifold coordinate $\lambda$ coincides (up to a factor of i) with the large radius coordinate $t$ measuring the size of the $\mathbb{P}^1$ in the geometry, but in general they are different, and related by a non-trivial symplectic transformation. We will see an example in the next section.

## 2.5 A more complicated example: local $\mathbb{P}^2$

In order to better understand the B model approach to topological string theory, it is useful to look at a rich example. An all-times favourite is the mirror to the local $\mathbb{P}^2$ CY manifold. Useful information on local $\mathbb{P}^2$ can be found in many papers, like e.g. [28, 52]. The corresponding mirror curve is given by

$$e^x + e^y + e^{-x-y} + \kappa = 0. \tag{2.45}$$

Here, $\kappa$ is a complex variable parametrizing the complex structure of the curve. It is also useful to introduce

$$z = \kappa^{-3}. \tag{2.46}$$

The curve (2.45) is in fact an elliptic curve in exponentiated variables.

**Exercise 2.5.** Use the transformation

$$
\begin{aligned}
e^x &= -\frac{\kappa}{2} + \frac{bY - a/2}{X + c}, \\
e^y &= \frac{a}{X + c},
\end{aligned}
\tag{2.47}
$$

to put the curve (2.45) in Weierstrass form,

$$Y^2 = 4X^3 - g_2 X - g_3. \tag{2.48}$$

Calculate $g_2$ and $g_3$, and verify that the discriminant of the curve is given by

$$\Delta(\kappa) = \frac{1 + 27\kappa^3}{\kappa}. \tag{2.49}$$

$\square$

The exercise above shows that there are three special points in the curve (2.45). The first one is $z = 0$, or $\kappa = \infty$. As we will see in a moment, this is the large radius point of the geometry, when the complexified Kähler parameter is large. The second special point is $z = \infty$, or $\kappa = 0$. This is the so-called *orbifold point*, where the theory can be described as a perturbed topological CFT. Finally, we have the point

$$z = -\frac{1}{27}, \tag{2.50}$$

where the discriminant (2.49) vanishes and the curve is singular. This is a conifold point, similar to the point $t = 0$ in the resolved conifold. The moduli space of local $\mathbb{P}^2$, with these three special

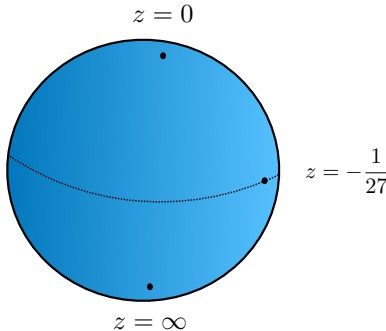

**Figure 2**: The moduli space of local $\mathbb{P}^2$, with the three special points $z = 0$ (large radius), $z = -1/27$ (conifold) and $z = \infty$ (orbifold).

points, is represented in Fig. 2 as a Riemann sphere. This sphere is divided in two hemispheres by the "equator" $|z| = 1/27$, which passes through the conifold point (2.50). The upper hemisphere, which includes the large radius point $z = 0$, can be regarded as the "geometric" phase of the model, where the topological string can be represented in terms of embedded Riemann surfaces. In the lower hemisphere, around the orbifold point $z = \infty$, the GW expansion around $z = 0$ does no longer converge, and one should use a more abstract picture in terms of a perturbed topological CFT coupled to gravity. See [53] for a discussion of the physics and mathematics of these moduli spaces.

The periods (2.36) are functions of $z$. The most convenient way to calculate these periods, as is well-known in mirror symmetry, is to find an ODE, or Picard–Fuchs (PF) equation, satisfied by all the periods. In the case of local $\mathbb{P}^2$, the PF equation reads [28]

$$\left(\theta^3 - 3z(3\theta + 2)(3\theta + 1)\theta\right) \Pi = 0, \tag{2.51}$$

where

$$\theta = z\frac{\mathrm{d}}{\mathrm{d}z} \tag{2.52}$$

and $\Pi$ is a period. This equation has three independent solutions, which can be calculated explicitly with the Frobenius method: a trivial, constant solution; a logarithmic solution $\varpi_1(z)$; and a double logarithmic solution $\varpi_2(z)$. If we introduce the power series,

$$
\begin{aligned}
\widetilde{\varpi}_1(z) &= \sum_{j \geq 1} 3\frac{(3j - 1)!}{(j!)^3}(-z)^j, \\
\widetilde{\varpi}_2(z) &= \sum_{j \geq 1} \frac{18}{j!}\frac{\Gamma(3j)}{\Gamma(1 + j)^2}\left\{\psi(3j) - \psi(j + 1)\right\}(-z)^j,
\end{aligned}
\tag{2.53}
$$

where $\psi(z)$ is the digamma function, we have

$$
\begin{aligned}
\varpi_1(z) &= \log(z) + \widetilde{\varpi}_1(z), \\
\varpi_2(z) &= \log^2(z) + 2\widetilde{\varpi}_1(z)\log(z) + \widetilde{\varpi}_2(z).
\end{aligned}
\tag{2.54}
$$

It is easy to see that the series in (2.53) have a radius of convergence $|z| = 1/27$, determined by the position of the conifold point. One can write $\widetilde{\varpi}_1(z)$ as a generalized hypergeometric function,

$$\widetilde{\varpi}_1(z) = -6z\,_4F_3\left(1, 1, \frac{4}{3}, \frac{5}{3}; 2, 2, 2; -27z\right). \tag{2.55}$$

We can now ask which combinations of the periods above lead to the complexified Kähler parameter and genus zero free energy determining the GW expansion (2.5). It turns out that the single logarithmic solution gives $t$, while the double logarithmic solution leads to the derivative of $F_0(t)$ (up to an overall factor). More precisely, we have

$$t = -\varpi_1(z), \qquad \partial_t F_0(t) = \frac{\varpi_2(z)}{6}. \tag{2.56}$$

Note that as $z \to 0$, we have $e^{-t} \sim z \to 0$, so this is the large radius limit, as we mentioned before. From (2.56) we can compute the genus zero free energy as

$$F_0(t) = \frac{t^3}{18} + 3e^{-t} - \frac{45}{8}e^{-2t} + \frac{244}{9}e^{-3t} - \frac{12333}{64}e^{-4t} + \mathcal{O}(e^{-5t}). \tag{2.57}$$

It can be seen from the PF equations that the convergence radius of the series (2.54) at $z = 0$ is set by the conifold singularity at $z = -1/27$. This means in particular that the radius of convergence of the genus zero free energy (2.57) is set by the value of $t$ at the conifold point [29, 45], which in this case is given by [54]

$$t\left(-\frac{1}{27}\right) = \frac{9}{\pi}\mathrm{Im}\left(\mathrm{Li}_2(e^{\pi i/3})\right) \pm \pi i, \tag{2.58}$$

where the choice of sign is due to the choice of branch cut of $\log(z)$. As noted above, it is expected that this radius of convergence is common to all genus $g$ free energies.

**Exercise 2.6.** Derive (2.57) from (2.53), (2.54) and (2.56). Verify the result with the topological vertex expansion in Exercise 2.3. □

The choice of periods in (2.56) defines the large radius frame. Let us now consider the conifold frame. In order to construct it, we have to find an appropriate conifold coordinate. This must be a combination of periods which vanishes at the conifold point and having good local properties there. In the case of local $\mathbb{P}^2$, this coordinate is given by

$$\lambda(z) = \frac{1}{4\pi}\left(\omega_c(z) - \pi^2\right), \tag{2.59}$$

where

$$\omega_c(z) = \log^2(-z) + 2\log(-z)\widetilde{\varpi}_1(z) + \widetilde{\varpi}_2(z). \tag{2.60}$$

The conifold frame is then defined by the periods

$$\begin{aligned}
\frac{\partial F_0^c}{\partial \lambda} &= -\frac{2\pi}{3}t \pm \frac{2\pi^2 i}{3}, \\
\lambda &= \frac{3}{2\pi}\partial_t F_0 \pm \frac{i}{2\pi}t - \frac{\pi}{2}.
\end{aligned} \tag{2.61}$$

The second relation is of course a consequence of (2.59). The choices of signs in (2.61) are correlated with the choice of branch cut of $\log(z)$ for $-1/27 < z < 0$, and they have been made in such a way that $\lambda$ and $\partial_\lambda F_0^c$ are real in that interval. The genus zero free energy in the conifold frame can be computed explicitly and it has the form,

$$F_0^c(\lambda) = \frac{1}{2}\lambda^2\left(\log\left(\frac{\lambda}{3^{5/2}}\right) - \frac{3}{2}\right) - \frac{\lambda^3}{36\sqrt{3}} + \frac{\lambda^4}{7776} + \frac{7\lambda^5}{87480\sqrt{3}} - \frac{529\lambda^6}{62985600} + \mathcal{O}\left(\lambda^7\right). \tag{2.62}$$

This is precisely the universal behavior obtained in (2.26) for the genus zero free energy (we have chosen the normalization of $\lambda$ so that it agrees with the canonical conifold coordinate, leading to the first equation in (2.26)). With some additional work, one can check that the higher genus free energies satisfy as well (2.26). As an example, the genus one and two free energies have the local expansion,

$$
\begin{aligned}
F_1^c(\lambda) &= -\frac{1}{12}\log(\lambda) + \frac{5\lambda}{72\sqrt{3}} - \frac{\lambda^2}{7776} - \frac{5\lambda^3}{17496\sqrt{3}} + \frac{283\lambda^4}{8398080} - \frac{43\lambda^5}{5668704\sqrt{3}} + \mathcal{O}\left(\lambda^6\right), \\
F_2^c(\lambda) &= -\frac{1}{240\lambda^2} + \frac{\lambda}{6480\sqrt{3}} - \frac{3187\lambda^2}{125971200} + \frac{239\lambda^3}{28343520\sqrt{3}} - \frac{19151\lambda^4}{28570268160} + \mathcal{O}\left(\lambda^5\right).
\end{aligned}
\tag{2.63}
$$

As we will see later, we will be able to recover these series (and more) from a non-perturbative definition of the topological string on local $\mathbb{P}^2$.

## 3 Resurgence and topological strings

Since the sequence of topological string free energies $F_g(\mathbf{t})$ is factorially divergent, one can have a first handle on non-perturbative aspects by using the theory of resurgence (a short review of this theory can be found in the Appendix). This program was first proposed in [47, 55, 56], and it has experienced many interesting developments in recent years. The first thing we can ask is: what is the *resurgent structure* of the topological string? The resurgent structure of a factorially divergent series, as we recall in the Appendix A, is the collection of trans-series and Stokes constants associated to the singularities of its Borel transform. Modulo some assumptions on endless analytic continuation of this transform, this is a well posed mathematical problem with a unique answer. The resurgent structure gives the collection of all non-perturbative sectors of the theory which can be obtained from the study of perturbation theory. From the point of view of physics, this gives candidate trans-series that complement the perturbative series and that can be used to obtain non-perturbative answers via Borel resummation. However, as we will see, the resurgent structure is interesting in itself, since the Stokes constants turn out to be, conjecturally, non-trivial invariants of the CY: the Donaldson–Thomas (DT) or BPS invariants.

### 3.1 Warm-up: the conifold and the resolved conifold

The problem of determining the resurgent structure of the topological string free energies is very difficult, due to the lack of explicit expressions for the genus $g$ free energies. There are however two cases where one can determine this structure analytically: the conifold free energies (2.26), i.e. the leading singularities of the topological string free energy near a conifold singularity, and the free energies of the resolved conifold (2.24). These two examples were first worked out by S. Pasquetti and R. Schiappa in [57] and they turn out to be fundamental ingredients in the full theory.

Let us start with the conifold free energies. We consider the formal power series

$$
\varphi(g_s) = \sum_{g \geq 2} b_{2g} g_s^{2g-2}, \qquad b_{2g} = \frac{B_{2g}}{2g(2g-2)}\lambda^{2-2g}.
\tag{3.1}
$$

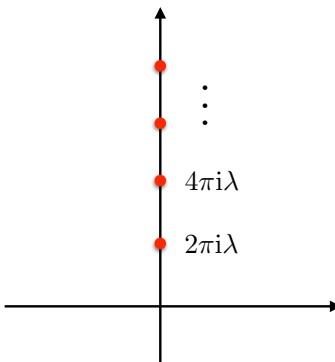

**Figure 3**: The singularities of the Borel transform (3.2) are located at non-zero integer multiples of $2\pi\mathrm{i}\lambda$. In the figure we show the Stokes ray through the singularities $2\pi\mathrm{i}\ell\lambda$ with $\ell \in \mathbb{Z}_{>0}$.

It turns out to be more convenient to use the Borel transform (A.2). One finds,

$$\widetilde{\varphi}(\zeta) = \sum_{g \geq 2} \frac{b_{2g}}{(2g-3)!} \zeta^{2g-3} = \sum_{g \geq 2} \frac{B_{2g}}{2g(2g-2)!} \lambda^{2-2g} \zeta^{2g-3}$$

$$= \frac{1}{\zeta} \left\{ -\frac{1}{12} + \frac{\lambda^2}{\zeta^2} - \frac{1}{4\sinh^2\left(\frac{\zeta}{2\lambda}\right)} \right\}. \tag{3.2}$$

(Note that the other version of the Borel transfom defined in (A.3), which is often used in the literature, is given by a primitive of this function, and that is why it is difficult to obtain an explicit expression for it.)

The singularities of the Borel transform (3.2) are located at

$$\zeta = 2\pi\mathrm{i}\ell\lambda, \qquad \ell \in \mathbb{Z}\backslash\{0\}, \tag{3.3}$$

see Fig. 3. They are double poles. Let us consider the Stokes ray going through the singularities with $\ell > 0$, at the angle $\theta = \pi/2$. The discontinuity of the lateral Borel resummations for that angle is simply computed by the sum of residues at those poles:

$$s_+(\varphi)(g_s) - s_-(\varphi)(g_s) = -2\pi\mathrm{i} \sum_{\ell=1}^{\infty} \mathrm{Res}_{\zeta=2\pi\mathrm{i}\ell\lambda} \left( \widetilde{\varphi}(\zeta) e^{-\zeta/g_s} \right)$$

$$= \frac{\mathrm{i}}{2\pi} \sum_{\ell \geq 1} \left\{ \frac{1}{\ell} \left( \frac{\mathcal{A}}{g_s} \right) + \frac{1}{\ell^2} \right\} e^{-\ell\mathcal{A}/g_s}, \tag{3.4}$$

where

$$\mathcal{A} = 2\pi\mathrm{i}\lambda. \tag{3.5}$$

If we consider the singularities in the negative imaginary axis, we find the same result, but with negative $\ell$. By comparing the second line in (3.4) to (A.15), we can read the trans-series:

$$\varphi_{\ell\mathcal{A}}(g_s) = \frac{1}{2\pi} \left\{ \frac{1}{\ell} \left( \frac{\mathcal{A}}{g_s} \right) + \frac{1}{\ell^2} \right\} e^{-\ell\mathcal{A}/g_s}, \qquad \ell \in \mathbb{Z}\backslash\{0\}. \tag{3.6}$$

From now on we will focus on the trans-series for positive $\ell$, although the results can be extended to the ones with negative $\ell$. We will normalize these trans-series as in (3.6), therefore in (3.4) the corresponding Stokes constant is one. In more complicated CY manifolds there are amplitudes of the form (3.6) with non-trivial Stokes constants, as we will see in a moment.

From a physicist perspective, the second line of (3.4) looks like a sum over multi-instantons with action $\mathcal{A}$. We will refer to (3.6) as a *Pasquetti–Schiappa $\ell$-instanton amplitude*. The expansion around each instanton is truncated at next-to-leading order in the coupling constant $g_s$. The sum over $\ell > 0$ can be performed in closed form and one finds,

$$(\mathfrak{S}_{\mathcal{A}} - 1)(\varphi) = \frac{\mathrm{i}}{2\pi} \left\{ \mathrm{Li}_2 \left( \mathrm{e}^{-\mathcal{A}/g_s} \right) - \frac{\mathcal{A}}{g_s} \log \left( 1 - \mathrm{e}^{-\mathcal{A}/g_s} \right) \right\} = \log \Phi_1 \left( -\frac{\mathcal{A}}{2\pi g_s} \right), \qquad (3.7)$$

where we have expressed the result in terms of the Stokes automorphism introduced in (A.18). This automorphism captures the discontinuity associated to all the singularities $\ell\mathcal{A}$, $\ell \in \mathbb{Z}_{>0}$, along the Stokes ray given by the imaginary axis. We have denoted it by $\mathfrak{S}_{\mathcal{A}}$. In the last step in (3.7) we have used (B.12) to identify this function as Faddeev's non-compact quantum dilogarithm $\Phi_{\mathsf{b}}(x)$, evaluated at $\mathsf{b} = 1$. Faddeev's quantum dilogarithm is a remarkable special function introduced in [58], which appears in many contexts in modern mathematical physics. In Appendix B we list some of its properties, which will be also useful in section 4. Since $\mathfrak{S}_{\mathcal{A}}$ is an automorphism, its action on $Z_{\mathrm{con}} = \mathrm{e}^{\varphi}$ is multiplicative:

$$\mathfrak{S}_{\mathcal{A}}(Z_{\mathrm{con}}) = \Phi_1 \left( -\frac{\mathcal{A}}{2\pi g_s} \right) Z_{\mathrm{con}}. \qquad (3.8)$$

**Remark 3.1.** Writing the Stokes automorphism in terms of Faddeev's quantum dilogarithm allows for a compact notation, but there is a deeper reason for it. In the local case, the topological string admits a deformation or refinement by using the so-called Omega background [43, 59]. This deformation can be parametrized by a complex number $\mathsf{b}$, and the undeformed or unrefined case corresponds to the value $\mathsf{b} = 1$. It turns out that the formula (3.7) admits a generalization to the refined a case, in which the r.h.s. involves Faddeev's dilogarithm for arbitrary $\mathsf{b}$; see [60] for the details.

As we mentioned above, the results for the conifold are very useful, and make it possible to obtain the trans-series for the resolved conifold immediately. The reason is that, thanks to (2.25), we can write the resolved conifold free energies as an infinite sum of conifold free energies:

$$F_g(t) = \sum_{m \in \mathbb{Z}} \frac{B_{2g}}{2g(2g-2)} \left( \mathrm{i}t + 2\pi m \right)^{2-2g}, \qquad g \geq 2. \qquad (3.9)$$

Therefore, the singularities of the Borel transform are of the form $\ell\mathcal{A}_m$, where $\ell \in \mathbb{Z}\backslash\{0\}$ and the "action" $\mathcal{A}_m$ is labelled by an additional integer $m$:

$$\mathcal{A}_m = 2\pi t + 4\pi^2 \mathrm{i}m, \qquad m \in \mathbb{Z}. \qquad (3.10)$$

The corresponding trans-series are also of the Pasquetti–Schiappa form. A plot of the very first singularities is shown in Fig. 4. Note that they are organized in infinite towers, and the singularities in each tower are obtained by changing the value of $m$ in (3.10). The Stokes rays going through the singularities $\ell\mathcal{A}_m$ for a fixed $m$ and $\ell \in \mathbb{Z}_{>0}$ accumulate along the imaginary axis. Such a pattern of Borel singularities is common in topological string theory on local CY

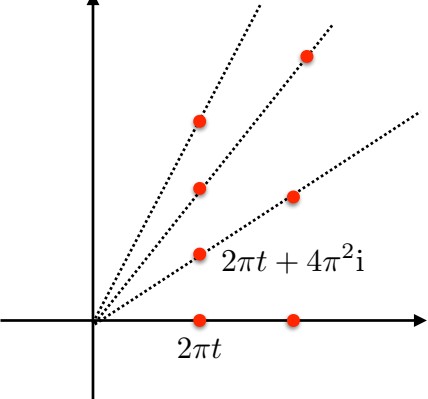
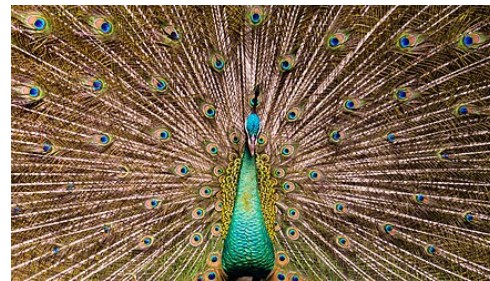

**Figure 4**: The singularities of the Borel transform in the case of the resolved conifold are located at non-zero integer multiples of $2\pi t + 4\pi^2 \mathrm{i} m$, where $m \in \mathbb{Z}$. As $m$ goes through $\mathbb{Z}$, the singularities form towers. The Stokes rays are somewhat similar to the feathers in a peacock's tail, hence the name "peacock patterns" for this structure of singularities (the picture on the right was made by Mallory Cessair and is courtesy of *Wikimedia Commons*).

manifolds, but also in complex Chern–Simons theory. Graphically, the set of Stokes rays going through Borel singularities is similar to the tail of a peacock, and for this reason these patterns were called "peacock patterns" in [61].

There is a simple extension of this result which is also useful. It was first discussed in [57, 62, 63] and further developed in [64]. Let us consider the expression (2.16) for $g \geq 2$, which is valid for the free energies in the large radius frame, and near the large radius point $\mathrm{Re}(t_i) \gg 1$. The first term in the second line is a sum of free energies for the resolved conifold, and it is easy to see that it is the only term growing factorially with $g$. Therefore, we expect that, close enough to the large radius point, we will have a sequence of Borel singularities at

$$\mathcal{A}_{\mathbf{d},m} = 2\pi \mathbf{d} \cdot \mathbf{t} + 4\pi^2 \mathrm{i} m, \qquad m \in \mathbb{Z}, \tag{3.11}$$

where $\mathbf{d}$ are the values of the degrees which lead to a non-zero GV invariant $n_0^{\mathbf{d}}$. The trans-series associated to these singularities are of the form

$$n_0^{\mathbf{d}} \, \varphi_{\ell \mathcal{A}_{\mathbf{d},m}}(g_s). \tag{3.12}$$

Therefore, $n_0^{\mathbf{d}}$ (which is an integer) has to be interpreted as the Stokes constant associated to the sequence of singularities (3.11), and the corresponding Stokes automorphism can be written as

$$\mathfrak{S}_{\mathcal{A}}(Z_{\mathrm{LR}}) = \left[ \Phi_1 \left( -\frac{\mathcal{A}}{2\pi g_s} \right) \right]^{n_0^{\mathbf{d}}} Z_{\mathrm{LR}}, \tag{3.13}$$

where $\mathcal{A}$ is given by (3.11), and we have denoted the partition function in the large radius frame by $Z_{\mathrm{LR}}$. Explicit numerical calculations show that the Borel singularities (3.11) indeed do occur, and their Stokes constants are given by the genus zero GV invariants [63, 64].

**Exercise 3.2.** Let us consider the constant map contribution (2.8). Show that it can be written as

$$c_g = -\frac{B_{2g}}{2g(2g-2)} \sum_{m=1}^{\infty} (2\pi m)^{2-2g} \tag{3.14}$$

Deduce that, if

$$\varphi_{\text{cm}}(g_s) = \sum_{g \geq 2} c_g g_s^{2g-2}, \tag{3.15}$$

one has the discontinuity formula

$$s_+(\varphi_{\text{cm}})(g_s) - s_-(\varphi_{\text{cm}})(g_s) = -\frac{\text{i}}{2\pi} \sum_{\ell \geq 1} \sigma(\ell) \left\{ \frac{1}{\ell} \left( \frac{\mathcal{A}}{g_s} \right) + \frac{1}{\ell^2} \right\} \text{e}^{-\ell \mathcal{A}/g_s}, \tag{3.16}$$

where

$$\mathcal{A} = 4\pi^2 \text{i}, \qquad \sigma(\ell) = \sum_{m|\ell} \left( \frac{\ell}{m} \right)^2. \tag{3.17}$$

This result was derived in [64] with a different technique. Conclude that

$$\mathfrak{S}_{\mathcal{A}} (Z_{\text{cm}}) = \prod_{\ell=1}^{\infty} \left[ \Phi_1 \left( -\frac{\ell \mathcal{A}}{2\pi g_s} \right) \right]^{-1} Z_{\text{cm}}, \tag{3.18}$$

where $Z_{\text{cm}} = \text{e}^{\varphi_{\text{cm}}}$.

$\square$

## 3.2 The general multi-instanton trans-series

In the examples of Stokes automorphisms considered so far, the location of the Borel singularities for the free energies or partition function has the following property: $\mathcal{A}$ is proportional to the flat coordinate of the frame in which we were computing the partition function, up to a shift by a constant. In addition, the Stokes automorphism acts multiplicatively on the partition function. In general, we expect $\mathcal{A}$ to be given by a linear combination of periods of the CY. In other words, and restricting ourselves to the local case with $g_{\Sigma} = 1$ for simplicity, we expect to have

$$\mathcal{A} = c \frac{\partial F_0}{\partial t} + dt + 4\pi^2 \text{i} m, \qquad m \in \mathbb{Z}. \tag{3.19}$$

This expectation was first stated in [65], based on previous insights on instantons in matrix models [66]. Additional arguments and evidence for this principle were given in [52, 67]. In addition it was emphasized in [64] that, with appropriate normalizations, Borel singularities are *integer* linear combinations of periods. This means that $c$ and $d$ in (3.19) are universal constants, times integers.

Let us now give a simple argument for obtaining the trans-series associated to a general Borel singularity, of the form (3.19). If $c = 0$, the instanton action is the flat coordinate $t$, up to a shift, and we expect the Stokes automorphism to act multiplicatively, i.e.

$$\mathfrak{S}_{\mathcal{A}}(Z(t)) = \exp \left[ \frac{\text{i}\mathsf{S}}{2\pi} \left( \text{Li}_2 \left( \text{e}^{-\mathcal{A}/g_s} \right) - \frac{\mathcal{A}}{g_s} \log \left( 1 - \text{e}^{-\mathcal{A}/g_s} \right) \right) \right] Z(t). \tag{3.20}$$

Here, $\mathsf{S}$ is a Stokes constant. We know from (3.7) that this formula is true for the Borel singularity at the conifold frame (3.5), with $\mathsf{S} = 1$, and for the Borel singularities (3.11) at large radius, with

$\mathsf{S} = n_0^{\mathbf{d}}$. It was conjectured in [64] that the formula (3.20) holds for arbitrary Borel singularities (3.19) with $c = 0$, and further tests were presented.

Let us now suppose that $c \neq 0$, and let us consider a frame where the flat coordinate $\tilde{t}$ is the instanton action $\mathcal{A}$ given in (3.19). This frame is called in [60, 64] an $\mathcal{A}$-frame, and is defined by the transformation

$$\begin{pmatrix} \partial_{\tilde{t}}\widetilde{F}_0 \\ \tilde{t} \end{pmatrix} = \begin{pmatrix} \alpha & \beta \\ c & d \end{pmatrix} \begin{pmatrix} \partial_t F_0 \\ t \end{pmatrix} + \begin{pmatrix} \sigma \\ 4\pi^2 \mathrm{i} m \end{pmatrix}, \tag{3.21}$$

where $\alpha$, $\beta$ are such that $\alpha d - \beta c = 1$, and $\sigma$ is a shift. We can now invert this transformation and use the general formula (2.43), to find

$$S(\tilde{t}, t) = -\frac{t\tilde{t}}{c} + \frac{d}{2c}t^2 + \frac{4\pi^2 \mathrm{i} m}{c}t + s(\tilde{t}), \tag{3.22}$$

where

$$s(\tilde{t}) = \frac{\alpha}{2c}\tilde{t}^2 + \left(\sigma - 4\pi^2 \mathrm{i} m\frac{\alpha}{c}\right)\tilde{t}. \tag{3.23}$$

The partition functions $Z(t; g_s)$, $\widetilde{Z}(\tilde{t}; g_s)$ are then related by (2.44). We note that we can write

$$Z(t; g_s) = \exp\left(-\frac{d}{2cg_s^2}t^2 - \frac{4\pi^2 \mathrm{i} m}{cg_s^2}t\right)\widehat{Z}(t; g_s), \tag{3.24}$$

where

$$\widehat{Z}(t; g_s) = \int \widetilde{Z}(\tilde{t}; g_s)\exp\left(\frac{t\tilde{t}}{cg_s^2} - \frac{s(\tilde{t})}{g_s^2}\right)\mathrm{d}\tilde{t}. \tag{3.25}$$

We will now assume that the Stokes automorphism acting on $Z(t; g_s)$ is obtained as a Fourier transform of the Stokes automorphism acting on $\widetilde{Z}(\tilde{t}; g_s)$. This is very natural, since the action of the Stokes automorphism can be regarded as a trans-series generalization of the perturbative partition function, and the Fourier transform acting on the perturbative sector extends naturally to the full trans-series. At the same time, since $\mathcal{A}$ is a flat coordinate in the $\tilde{t}$ frame, the action of the Stokes automorphism on $\widetilde{Z}(\tilde{t}; g_s)$ is multiplicative, according to the conjecture explained above. We conclude that

$$\begin{aligned} \mathfrak{S}_{\mathcal{A}}(Z(t; g_s)) &= \int \mathfrak{S}_{\mathcal{A}}(\widetilde{Z}(\tilde{t}; g_s))\mathrm{e}^{-S(t,\tilde{t})/g_s^2}\mathrm{d}\tilde{t} \\ &= \exp\left(-\frac{d}{2cg_s^2}t^2 - \frac{4\pi^2 \mathrm{i} m}{cg_s^2}t\right)\left[\Phi_1\left(-\frac{g_s c}{2\pi}\partial_t\right)\right]^{\mathsf{S}}\widehat{Z}(t; g_s). \end{aligned} \tag{3.26}$$

We have used here the standard property that insertions of $\tilde{t}$ inside the Fourier transform can be traded by derivatives. We can also write this as

$$\mathfrak{S}_{\mathcal{A}}(\widehat{Z}(t; g_s)) = \left[\Phi_1\left(-\frac{g_s c}{2\pi}\partial_t\right)\right]^{\mathsf{S}}\widehat{Z}(t; g_s). \tag{3.27}$$

This formula was derived in [64, 68, 69] by using a more complicated method, based on the holomorphic anomaly equations of [24], which has the advantage that it applies to compact CY manifolds as well. The derivation presented here, in a slightly less general form, can be found in [70].

We can write (3.27) more explicitly as

$$\mathfrak{S}_{\mathcal{A}}(\widehat{Z}(t; g_s)) = \exp\left[\frac{\mathrm{i}\mathsf{S}}{2\pi}\left(\mathrm{Li}_2\left(\mathcal{C}\mathrm{e}^{-cg_s\partial_t}\right) - cg_s\partial_t\log\left(1 - \mathcal{C}\mathrm{e}^{-cg_s\partial_t}\right)\right)\right]\widehat{Z}(t; g_s), \tag{3.28}$$

where we have introduced a parameter $\mathcal{C}$ to keep track explicitly of the exponentially small corrections. By expanding the r.h.s. of this equation in powers of $\mathcal{C}$ we find

$$\widehat{Z}(t;g_s) + \frac{\mathrm{iS}}{2\pi}\mathcal{C}\left(1 + cg_s\partial_t\widehat{F}(t - cg_s;g_s)\right)\mathrm{e}^{\widehat{F}(t-cg_s;g_s)} + \mathcal{O}(\mathcal{C}^2). \tag{3.29}$$

The action of the Stokes automorphism on the free energy follows from

$$\mathfrak{S}_{\mathcal{A}}(\widehat{F}(t;g_s)) = \log\mathfrak{S}_{\mathcal{A}}(\widehat{Z}(t;g_s)), \tag{3.30}$$

where $\widehat{F}(t;g_s) = \log\widehat{Z}(t;g_s)$. We can introduce the multi-instanton sectors of the free energy as

$$\mathfrak{S}_{\mathcal{A}}(\widehat{F}(t;g_s)) - \widehat{F}(t;g_s) = \mathrm{i}\sum_{\ell\geq 1}\mathcal{C}^\ell F^{(\ell)}(t;g_s), \tag{3.31}$$

and we find that the first instanton sector is given by

$$F^{(1)}(t;g_s) = \frac{\mathsf{S}}{2\pi}\left(1 + cg_s\partial_t\widehat{F}(t - cg_s;g_s)\right)\mathrm{e}^{\widehat{F}(t-cg_s;g_s)-\widehat{F}(t;g_s)}. \tag{3.32}$$

Higher instanton sectors can be obtained in a straightforward way.

Let us make some comments on the structure of the formula (3.32). First of all, the total free energy $\widehat{F}(t;g_s)$ differs from $F(t;g_s)$ only in its genus zero piece, i.e. we have

$$\widehat{F}_0(t) = F_0(t) + \frac{d}{2c}t^2 + \frac{4\pi^2\mathrm{i}m}{c}t, \tag{3.33}$$

and is such that

$$\mathcal{A} = c\frac{\partial\widehat{F}_0(t)}{\partial t}. \tag{3.34}$$

In particular, the exponential factor in (3.32) has the $g_s$ expansion

$$\exp\left(\widehat{F}(t - cg_s;g_s) - \widehat{F}(t;g_s)\right) = \mathrm{e}^{-\mathcal{A}/g_s}\left(1 + \mathcal{O}(g_s)\right), \tag{3.35}$$

so (3.32) is manifestly a non-perturbative correction. The exponent in (3.35) can be interpreted as the difference between the free energies of two different backgrounds, or points in the moduli space of the CY: the background $t$, and the background $t - cg_s$, in which $t$ is shifted. It is easy to see that in the $\ell$-th instanton sector the shift is given by $t - \ell cg_s$. Since, with the appropriate normalizations, $c$ is an integer, this suggests that $t$ is "quantized" in units of $g_s$. Such a quantization is typical of topological string theories with large $N$ duals, in which the CY modulus is interpreted as a 't Hooft parameter and has the form $Ng_s$, where $N$ is the rank of the matrix model [9, 11, 13]. We will elaborate on this in section 4.

The result that we have derived for the Stokes automorphism is valid for local CY manifolds with a single modulus, but it has an obvious generalization to general CYs [64, 69]. In this case, it is more convenient to use the projective free energies

$$\mathfrak{F}_g(X^I) = \left(X^0\right)^{2-2g}F_g(\mathbf{t}), \tag{3.36}$$

generalizing the projective prepotential (2.32). Let us introduce a charge vector $\boldsymbol{\gamma} = (c^I, d_I)$, where $I = 0, 1, \cdots, s$, which generalizes the numbers $c, d, m$ appearing in (3.19). The location of a Borel singularity is given by

$$\mathcal{A}_{\boldsymbol{\gamma}} = c^I\mathcal{F}_I + d_I X^I, \tag{3.37}$$

where summation over the repeated indices is understood. If all $c^I = 0$, the Stokes automorphism is given by the formula (3.20). If not all the $c^I$s vanish, one first defines a new genus zero free energy by

$$\mathcal{A}_\gamma = c^I \frac{\partial \widehat{\mathfrak{F}}_0}{\partial X^I}, \tag{3.38}$$

as in the local case. It can be written as

$$\widehat{\mathfrak{F}}_0(X^I) = \mathfrak{F}_0(X^I) + \frac{1}{2} a_{IJ} X^I X^J, \qquad a_{IJ} c^I = d_J, \tag{3.39}$$

which is the counterpart of (3.33) (the final formulae will not depend on the choice of $a_{IJ}$, but only on $c^I$, $d_J$.) One also has to define a new genus one free energy

$$\widehat{\mathfrak{F}}_1(X^I) = \mathfrak{F}_1(X^I) - \left(\frac{\chi}{24} - 1\right) \log X^0, \tag{3.40}$$

where we recall that $\chi$ is the Euler characteristic of the CY $M$. This is a new ingredient in the compact case which was found in [64]. The redefinitions of the genus zero and one free energies lead to a new total free energy which will be denoted by $\widehat{\mathfrak{F}}(X^I; g_s)$. It is given by

$$\widehat{\mathfrak{F}}(X^I; g_s) = g_s^{-2} \widehat{\mathfrak{F}}_0(X^I) + \widehat{\mathfrak{F}}_1(X^I) + \sum_{g \geq 2} g_s^{2g-2} \mathfrak{F}_g(X^I). \tag{3.41}$$

Then, one has the following generalization of (3.27),

$$\mathfrak{S}_{\mathcal{A}_\gamma}(\widehat{Z}) = \left[\Phi_1\left(-\frac{g_s c^I \partial_I}{2\pi}\right)\right]^{\mathsf{S}_\gamma} \widehat{Z}, \tag{3.42}$$

where we have denoted $\widehat{Z} = \mathrm{e}^{\widehat{\mathfrak{F}}}$, $\mathsf{S}_\gamma$ is the Stokes constant corresponding to the ray of singularities $\ell \mathcal{A}_\gamma$, $\ell \in \mathbb{Z}_{>0}$, and

$$\partial_I = \frac{\partial}{\partial X^I}. \tag{3.43}$$

It is also useful to consider the *dual* partition function and the action of the Stokes automorphism (3.42) on it. The dual partition function is a discrete Fourier transform of the usual partition function, and it diagonalizes the operator action appearing in (3.42). It depends on an additional set of variables $\rho_I$, $I = 0, 1, \ldots, n$, and it is given by

$$\tau\left(X^I, \rho_I; g_s\right) = \mathrm{e}^{\frac{1}{2g_s^2} X^I \rho_I} \sum_{\boldsymbol{\ell} \in \mathbb{Z}^n} \mathrm{e}^{\kappa \rho_I \ell^I / g_s} Z\left(X^I + \ell^I g_s; g_s\right). \tag{3.44}$$

Let us now introduce the following quantity, associated to a charge vector $\boldsymbol{\gamma}$

$$\boldsymbol{X}_\gamma = \sigma(\boldsymbol{\gamma}) \exp\left[-\kappa g_s^{-1}\left(d_I X^I - \rho_I c^I\right)\right], \tag{3.45}$$

where $\sigma(\boldsymbol{\gamma}) = (-1)^{d_I c^I}$. It is an easy exercise to show that the action of the Stokes automorphism on this function is given by [69]

$$\mathfrak{S}_{\mathcal{A}_\gamma}\left(\tau\left(X^I, \rho_I; g_s\right)\right)$$
$$= \exp\left(\frac{\mathrm{i}\mathsf{S}_\gamma}{2\pi} L_{\sigma(\gamma)}(\boldsymbol{X}_\gamma)\right) \tau\left(X^I - \frac{\mathrm{i}\mathsf{S}_\gamma}{2\pi} g_s c^I \log(1 - \boldsymbol{X}_\gamma), \rho_I - \frac{\mathrm{i}\mathsf{S}_\gamma}{2\pi} g_s d_I \log(1 - \boldsymbol{X}_\gamma); g_s\right), \tag{3.46}$$

where $L_\epsilon(z)$ is the twisted Rogers dilogarithm

$$L_\epsilon(z) = \mathrm{Li}_2(z) + \frac{1}{2} \log\left(\epsilon^{-1}z\right) \log(1-z). \tag{3.47}$$

The formula (3.46) agrees with the wall-crossing formula obtained in a very different context in [71]. A remarkable aspect of (3.46) is that it induces a shift on the coordinates $X^I, \rho_I$ of the dual partition function. It is easily seen that this can be written as a transformation acting on a $\boldsymbol{X_{\gamma'}}$ of the form

$$\boldsymbol{X_{\gamma'}} \to \boldsymbol{X_{\gamma'}} \left(1 - \boldsymbol{X_\gamma}\right)^{-\frac{\mathrm{iS}_\gamma}{2\pi}\langle\boldsymbol{\gamma},\boldsymbol{\gamma'}\rangle}, \tag{3.48}$$

where

$$\langle\boldsymbol{\gamma},\boldsymbol{\gamma'}\rangle = d_I c'^I - c^I d'_I \tag{3.49}$$

is the symplectic pairing between the two charge vectors. The equation (3.48) describes the transformation of quantum periods under a Stokes automorphism, and it is known in that context as the Delabaere–Dillinger–Pham (DDP) formula [72, 73], see e.g. [74] for recent developments and references to previous literature (the DDP transformation is also known as a cluster transformation, or a Kontsevich–Soibelman symplectomorphism, depending on the context).

**Remark 3.3.** We have not been careful about the normalization of the charge vector $\boldsymbol{\gamma}$, but it can be seen that $(2\pi\mathrm{i})^{-1}\langle\boldsymbol{\gamma},\boldsymbol{\gamma'}\rangle$, which appears in the exponent of (3.48), is an integer; see [69] for details.

The formulae above for the instanton amplitudes and Stokes automorphism -(3.32), (3.42), and (3.46)- have a wide range of applications. They can be derived from the holomorphic anomaly equations of [24] and the conifold behavior (2.26) at the singular loci of moduli space [64, 68], therefore they apply to topological strings on arbitrary CY threefolds. Since the free energies obtained from topological recursion satisfy the holomorphic anomaly equations when the spectral curve has genus greater or equal to one [49], it follows that their resurgent structure is governed by the formulae above, provided they exhibit conifold behavior. This is in particular the case for multi-cut Hermitian matrix models, and as shown in [70, 75], the expressions (3.32), (3.42) give the general form of large $N$ instantons in matrix models, generalizing the one-cut case worked out in [55, 76]. It is important to note that the formulae (3.32), (3.42), are testable, since according to elementary resurgence results (reviewed in Appendix A), the large genus behaviour of the sequence of free energies $F_g$ is governed by the instanton with the smallest action (in absolute value). This has been verified in detail in many examples, starting from the work [52, 67] (where the very first terms of the $g_s$ expansion of (3.32) were first found), and more recently in [64, 68]. The formula (3.46) can be checked independently [69] in the case of the dual partition function obtained from topological recursion in [77], which is a formal tau function of the Painlevé I equation (this has been reviewed in the lectures by K. Iwaki in this school). Related results in the case of supersymmetric gauge theory partition functions have been obtained in [78].

## 3.3 BPS states and the resurgent structure of topological strings

In the last section we have derived general results for the trans-series associated to the different Borel singularities, but we still need to know the precise location of the singularities and the corresponding Stokes constants.

The positions of the Borel singularities for the topological string free energies depend on the value of the moduli of the CY. As we move in moduli space, the singularities change their

position and sometimes change *discontinuously*. This phenomenon was first observed in [72], in the resurgent structure of quantum or WKB periods associated to the Schrödinger equation. This discontinuous change will be referred to as *wall-crossing*, since it is indeed related to wall-crossing phenomena for BPS states in supersymmetric gauge theory [79–81], as reviewed in A. Neitzke lectures, and in the theory of Donaldson–Thomas invariants [82].

In the theory of BPS states or Donaldson–Thomas invariants on a CY threefold, the BPS states are characterized by a charge $\gamma \in \Gamma$, where $\Gamma$ is an appropriate lattice. For example, for a compact CY threefold $M$ in the A-model one has

$$\Gamma = H^{\mathrm{ev}}(M, \mathbb{Z}), \tag{3.50}$$

and its rank is $2(s + 1)$, where we recall that $s = h^{1,2}(M)$. If we choose a basis for this lattice, we can write $\gamma$ in terms of two pairs of vectors of rank $s + 1$, with entries, $\gamma = (c^I, d_I)$, where $I = 0, 1, \cdots, s$. In the context of type IIA superstring theory, the BPS states are obtained by wrapping a D2$p$ brane around a cycle of even dimension $2p$ inside the CY threefold, leading to a four-dimensional BPS particle in the uncompactified directions. We can think of $c^0$, $d_0$ as D6 and D0 brane charges, respectively, and of $c^a$, $d_a$, $a = 1, \cdots, s$, as D4 and D2 brane charges, respectively. The central charge corresponding to such an element of $\Gamma$ is given by

$$Z_\gamma = c^I \mathcal{F}_I + d_I X^I, \tag{3.51}$$

where summation over the repeated indices is understood. Let us note that, in the case of toric CY manifolds, D6 branes decouple, and the charge $\gamma$ is specified by $2s + 1$ integers which we will denote by $c^a$, $d_a$ and $m$, with $a = 1, \cdots, s$. The central charge reads then

$$Z_\gamma = c^a \frac{\partial F_0}{\partial t_a} + d_a t_a + 4\pi^2 \mathrm{i} m. \tag{3.52}$$

There is of course a B model, mirror description of BPS states on the mirror manifold $M^\star$ (or, physically, in the type IIB superstring compactified on $M^\star$) in which the lattice is $\Gamma = H^3(M^\star, \mathbb{Z})$. Given a point in moduli space, one can define BPS or DT invariants associated to a charge $\gamma$, which we will denote by $\Omega_\gamma$. The spectrum of BPS states is the set of charges $\gamma$ for which $\Omega_\gamma \neq 0$. The invariants $\Omega_\gamma$ (and therefore the spectrum of BPS states) can jump discontinuously as we move in moduli space, and this is the phenomenon of wall-crossing.

**Example 3.4.** A simple example of BPS spectrum and invariants occurs in Seiberg–Witten (SW) theory [79]. In the CY setting, this example arises as a B model description of the BPS states on the local CY manifold described by the so-called SW curve

$$y^2 + 2\cosh(x) = 2u. \tag{3.53}$$

We note that the variable $x$ appears in exponentiated form, but not the variable $y$. The moduli space is parametrized by the complex number $u$ (this is the famous $u$-plane of SW theory), and there are two independent periods which can be chosen to be

$$\begin{aligned} a(u) &= \frac{2\sqrt{2}}{\pi}\sqrt{u+1}\, E\left(\frac{2}{u+1}\right), \\ a_D(u) &= \frac{\mathrm{i}}{2}(u-1)\,_2F_1\left(\frac{1}{1}, \frac{1}{2}, 2; \frac{1-u}{2}\right). \end{aligned} \tag{3.54}$$

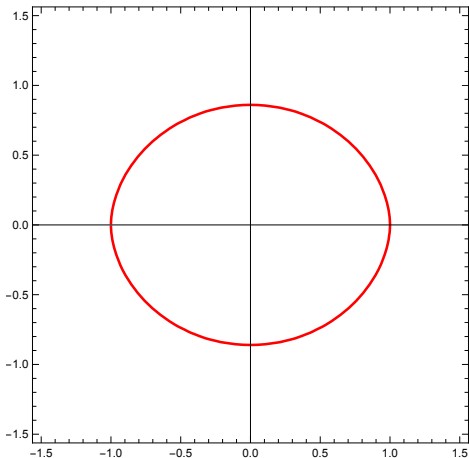

**Figure 5**: The curve of marginal stability in the $u$-plane, defined by the equation (3.57).

The lattice of charges here has rank two, and we will write a charge $\boldsymbol{\gamma}$ as

$$\boldsymbol{\gamma} = (\gamma_e, \gamma_m), \tag{3.55}$$

where $\gamma_{e,m}$ refers to the electric (respectively, magnetic) charge. Then, the central charge is given by

$$Z_{\boldsymbol{\gamma}}(u) = 2\pi \left( \gamma_e a(u) + \gamma_m a_D(u) \right), \tag{3.56}$$

where we have introduced an appropriate normalization factor $2\pi$ for the periods. The spectrum of BPS states in this theory has been investigated intensively, see e.g. [79–81, 83], and has been described in detail in the lectures by A. Neitzke in this school. First of all, the spectrum depends on the value of the modulus $u$. Inside the so-called *curve of marginal stability*, defined by

$$\mathrm{Im}\left( \frac{a_D}{a} \right) = 0, \tag{3.57}$$

we have the so-called strong coupling spectrum: the only stable states have charges

$$\boldsymbol{\gamma}_M = (0, 1), \qquad \boldsymbol{\gamma}_D = (1, 1), \tag{3.58}$$

corresponding to a magnetic monopole and a dyon, respectively. Outside this curve, we have the so-called weak coupling spectrum, consisting of a $W$-boson and a tower of dyons, with charges, respectively,

$$\boldsymbol{\gamma}_W = (1, 0), \qquad \boldsymbol{\gamma}_n = (n, 1), \quad n \in \mathbb{Z}. \tag{3.59}$$

See Fig. 5 for a plot of the curve of marginal stability in the $u$-plane. States with charges $-\boldsymbol{\gamma}$ also belong to the spectrum. The corresponding DT invariants have the values

$$\Omega_{(0,1)} = \Omega_{(1,1)} = 1, \tag{3.60}$$

in the strong coupling region, while in the weak coupling region we have

$$\Omega_{(n,1)} = 1, \qquad \Omega_{(1,0)} = -2. \tag{3.61}$$

$\square$

Let us now reconsider the information obtained on the resurgent structure of the topological string, in section 3.1, in the light of the theory of BPS states and invariants. For the free energies in the large radius frame, and near the large radius point, the resurgent structure includes Borel singularities at the positions (3.11). These can be identified with central charges of BPS states due to D0-D2 branes, with charges $\boldsymbol{\gamma} = (0, \cdots, 0, d_I)$, where $I = 0, 1, \cdots, s$ and $d_0 = m$ is identified with the D0 charge. Their Stokes constants are given by the GV invariants

$$n_0^{\mathbf{d}} = \Omega_{\boldsymbol{\gamma}=(0,\cdots,0,d_I)}, \qquad \mathbf{d} = (d_1, \cdots, d_s), \tag{3.62}$$

and as indicated in (3.62) they can be identified with the DT invariant for a D0-D2 BPS state (see e.g. [84]). In addition, for the free energies in the conifold frame, the resurgent structure near the conifold locus includes a BPS state which becomes massless at the conifold point, with DT invariant equal to 1. This is also expected from the work [85, 86], where it was pointed out that the conifold behavior of the free energies is due to a single BPS hypermultiplet becoming massless at the conifold locus. In the local case this is typically a D4 state, while in the compact case (e.g. in the quintic CY) it is often a D6 state.

Additional evidence for the connection between Borel singularities and BPS states comes from the constant map contribution to the topological string free energies in (2.7). As shown in Exercise 3.2, this contribution leads to Borel singularities at

$$\mathcal{A}_\ell = 4\pi^2 i\ell, \qquad \ell \in \mathbb{Z}_{>0}, \tag{3.63}$$

which can be identified with the central charge of a bound state of $\ell$ D0-branes. The corresponding charge vector is $\boldsymbol{\gamma}_\ell = (0, \cdots, 0, \ell, 0, \cdots, 0)$. From (3.18) one can read that the Stokes constant is given by

$$\mathsf{S}_{\boldsymbol{\gamma}_\ell} = -\chi = \Omega_{\boldsymbol{\gamma}_\ell}, \tag{3.64}$$

for all $\ell \in \mathbb{Z}_{>0}$. In this formula we have included the prefactor $\chi$ multiplying the constant map contribution in (2.7). This result was obtained in [94] (in that paper they considered the resolved conifold with $\chi = 2$ but of course their result generalizes trivially to any CY). As noted in (3.64) the value of the Stokes constant agrees again with the result for the DT invariant of $\ell$ D0-branes.

These results give evidence that the Borel singularities at a given point of moduli space can be identified with central charges of BPS states at that point. Moreover, the Stokes constants have to be identified with BPS or DT invariants. This can be formulated as the following

**Conjecture 3.5.** The resurgent structure of the topological string free energy can be characterized as follows:

1. The total topological string free energy in a given frame is a resurgent function. Its Borel singularities are integer linear combinations of the CY periods, as in (3.37). These singularities are determined by a charge vector $\boldsymbol{\gamma}$, and their location is given by the central charge (3.51) of a BPS state with the same charge vector (up to a normalization).

2. The singularities display a multi-covering structure: given a singularity $\mathcal{A}_{\boldsymbol{\gamma}}$, all its integer multiples $\ell\mathcal{A}_{\boldsymbol{\gamma}}$, $\ell \in \mathbb{Z}\backslash\{0\}$, appear as singularities as well. The Stokes automorphism for the singularities occurring along a half-ray $\ell\mathcal{A}_{\boldsymbol{\gamma}}$, $\ell \in \mathbb{Z}_{>0}$ is given by (3.20) (for the case in which all $c^I = 0$) or (3.42) (for the case in which not all $c^I$ vanish).

3. The Stokes constant $\mathsf{S}_{\boldsymbol{\gamma}}$ appearing in these Stokes automorphisms is the BPS or DT invariant $\Omega_{\boldsymbol{\gamma}}$ associated to the BPS state with central charge $\mathcal{A}_{\boldsymbol{\gamma}}$.

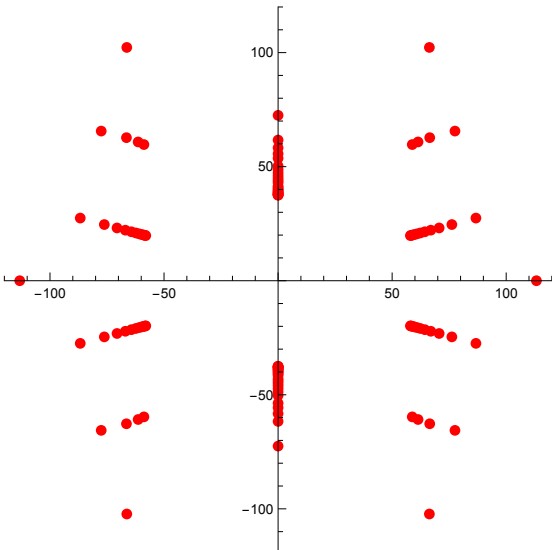

**Figure 6**: A numerical approximation to the Borel singularities of the free energies $F_g(t)$ in the large radius frame, for local $\mathbb{P}^2$, and for $z = -10^{-4}$. To obtain this approximation, we first consider the sequence of free energies up to $g = 120$, and then we determine and plot the poles of the diagonal Padé approximant to its Borel transform. The singularity in the positive imaginary axis corresponds to (3.5) and is due to the D4 state that becomes massless at the conifold. It occurs at $2\pi i \lambda \approx 37.5i$. Since $2\pi t \approx 57.9 + 2\pi^2 i$, the other singularities in the plot correspond to the tower (3.11) with $d = 1$, $m = 0, 1, -2$. As expected, singularities appear in pairs $\mathcal{A}, -\mathcal{A}$.

This conjecture concerns the resurgent structure of the topological string free energies in a fixed frame. However, it follows from the description that the Borel singularities and Stokes constant do *not* depend on the frame. Let us clarify this point. The free energies in a given frame $F_g$ are not globally defined on the moduli space, and they are analytic only on a region, typically centered around a special point, like the large radius point or the conifold point. At a point in the moduli space where the free energies in two different frames are well-defined, they have conjecturally the *same* resurgent structure, determined by the BPS structure at that point. However, the form of the Stokes automorphism might be different, depending on whether the frame is an $\mathcal{A}$-frame or not. As an example of this, let us consider the Borel singularity associated to the massless BPS state at the conifold point, which appears in the resurgent structure of the free energies in the conifold frame. According to the above, it should be also present in other frames, like e.g. the large radius frame, and indeed there is ample numerical evidence that this is the case [52, 63, 64, 68].

The invariance of the Borel singularities and Stokes constants under a change of frame is natural from the point of view of the generalized Fourier transform relating different frames: we can gather the information on the resurgent structure in a trans-series, i.e. in a collection of non-perturbative corrections to the perturbative partition function. The Stokes constants are coefficients in this trans-series, and the location of the Borel singularities can be read from the exponentially small terms in $g_s$. Under a change of frame, the full trans-series transforms under a generalized Fourier transform. This does not change the coefficients of the trans-series, nor the instanton actions appearing in the exponentially small terms in $g_s$, as we saw in e.g. (3.35).

As an illustration of the above description, we show in Fig. 6 a numerical calculation of the

Borel singularities for the free energies of local $\mathbb{P}^2$ in the large radius frame, for $z = -10^{-4}$, which is rather near the large radius point $z = 0$. As expected, we can see the very first singularities in the tower (3.11) due to D2-D0 BPS bound states, with $2\pi t \approx 57.9 + 2\pi^2 i$. We can also see the singularity (3.5) in the imaginary axis at $2\pi i \lambda \approx 37.5i$, which is due to the D4 BPS state which becomes massless at the conifold. The instanton associated to this singularity governs the large order behavior of the genus $g$ free energies for a wide range of values of $z$ in the "geometric" phase $|z| < 1/27$, including the value shown in Fig. 6. We note that this numerical calculation detects only the singularities which are closer to the origin. In order to see more singularities, one has to use more terms in the series and more sophisticated numerical techniques.

One of the ingredients of conjecture 3.5 is that the non-perturbative sectors of the topological string are associated to BPS states which are obtained by wrapping D-branes around cycles in the CY manifold. The role of D-branes in providing a source for exponentially small non-perturbative effects in the string coupling constant was already emphasized in [87], and it was verified explicitly in the context of non-critical strings [88, 89], where non-perturbative effects can be obtained via a resurgent analysis of the string equations in double-scaled matrix models [90]. The conjecture 3.5 extends this picture to topological strings on CY threefolds.

This conjecture 3.5 was built up in various works. The construction of explicit trans-series was started in [52, 57, 67]. General explicit formulae for the local case and the general case were obtained in [64, 68], respectively. A compact formula for the Stokes automorphism based on these developments was worked out in [69]. The connection between Stokes constants and BPS invariants was anticipated in [91]. A first formulation of the conjecture (in a special limit) was proposed in [3], stimulated by a similar connection discovered in complex Chern–Simons theory in [61, 92, 93]. The conjecture was shown to hold for the resolved conifold in [94–96]. The general formulation above can be found in [60, 69, 70]. In [60, 97] the conjecture is generalized to the refined topological string and to the real topological string, respectively.

There is both direct and indirect evidence for the conjecture 3.5. Important indirect evidence for the conjecture comes from comparison with a different line of work, studying the geometry of the hypermultiplet moduli space in CY compactifications (see [84] for a review). It was found in [71, 98] that there is a natural action of the so-called Kontsevich–Soibelman automorphisms on the topological string partition function, which involves the DT invariants of the CY. As noted in [69], this action turns out to be identical to the Stokes automorphism that we have just described, e.g. in (3.46), *provided* the Stokes constants are identified with DT invariants.

There is additional indirect evidence for the conjecture 3.5 for local CY manifolds. In this case one can consider WKB, or quantum periods associated to the quantum version of the curve (2.34), in which $x, y$ are promoted to Heisenberg operators (we will come back to this subject in section 4). These periods define a different topological string theory, usually called the *Nekrasov–Shatashvili* (NS) topological string [7]. The quantum periods associated to the quantum mirror curve are also factorially divergent power series, and one can study their resurgent structure (see e.g. [74] for references to the extensive literature on the subject). In [99] it was argued that, in the local case, the Stokes constants appearing in the resurgent structure of the standard topological string are the same ones appearing in the resurgent structure of the quantum periods. The latter should be directly related to DT invariants, as expected from the 4d results of [100].

More evidence for the conjecture comes from direct comparisons between calculations of Stokes constants, and calculations of BPS invariants. As we saw above, one of the simplest examples of a BPS spectrum and invariants is the one appearing in SW theory, which displays already a non trivial wall-crossing structure. One can associate to this theory a sequence of topological string free energies $\mathcal{F}_g$, $g \geq 0$, in many different ways, e.g. by considering topological

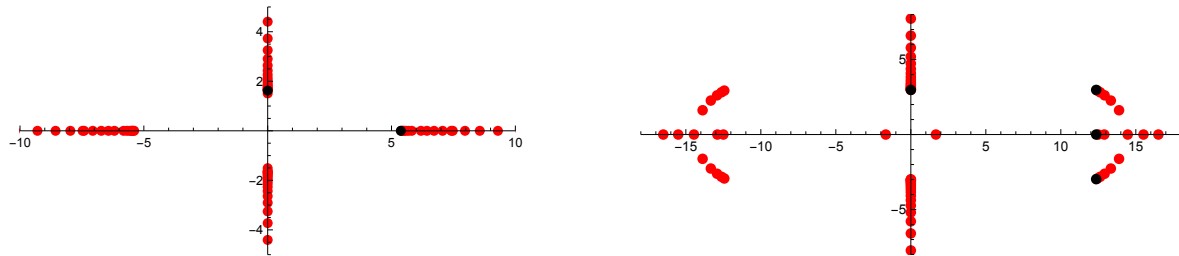

**Figure 7**: The Borel singularities of $\mathcal{F}(t)$ in SW theory, inside the curve of marginal stability with $u = 1/2$ (left), and outside the curve with $u = 5/2$ (right). The black dots in the figure in the left occur at the values $2\pi a_D$, $2\pi(a + a_D)$, and corresponds to the monopole and dyon state, respectively. The black dots in the figure in the right occur at the value $2\pi a$ (on the positive real axis), corresponding to the $W$-boson, at the value $2\pi a_D$ (on the imaginary axis), corresponding to the manopole, and at the values $2\pi(a \pm a_D)$ (above and below the positive real axis, respectively). The latter correspond to the dyons $(1, \pm 1)$, which are the first states in the tower.

recursion as applied to the SW curve (3.53). Can we match the BPS structure to the resurgent structure of these topological string free energies? This was answered in the affirmative in [70], by a numerical study of the sequence of free energies $\mathcal{F}_g$. It is convenient to do this in the so-called magnetic frame, in which the flat coordinate is chosen to be $t = -ia_D(u)$ (we recall that $a_D(u)$ is defined in (3.54)). If the conjecture 3.5 is correct, we should find a very different structure of singularities in the Borel plane depending on whether $u$ is inside or outside the curve of marginal stability. Inside the curve, we should find two Borel singularities (together with their reflections), corresponding to the monopole and dyon states. Outside the curve, we should find the monopole, the $W$ particle, and a tower of dyons. This is precisely what is obtained in a numerical analysis, as shown in Fig. 7. Since this is a numerical approximation, we only see the very first dyons in the tower (i.e. the ones with lowest masses). In addition, a detailed numerical calculation in [70] confirms the values of the DT invariants (3.60), (3.61).

We conclude then that the resurgent structure of the topological string is governed by the DT theory of the CY, and in particular gives a new perspective on the DT invariants and their wall-crossing. We should point out that the existence of some sort of relation between general BPS invariants and the topological string free energies has been suspected for a long time. It features for example in the so-called OSV conjecture [101], which equates the BPS invariants with a certain integral involving the topological string partition function. However, since this integral is not well-defined it is difficult to make sense of the OSV conjecture. In particular, it is not clear why the integral of the topological string partition function should undergo wall-crossing. The conjecture 3.5 is in contrast well-defined and can be tested. One of the key insights in the conjecture 3.5 is to relate the BPS invariants to the resurgent structure of the topological string, which *does* undergo wall-crossing, as it has been known in related examples since the work of [72].

## 4   Topological strings from quantum mechanics

In this final section we will explore the question of finding a non-perturbative completion of the topological string, i.e. of finding a well-defined function whose asymptotic expansion reproduces the perturbative free energy as its asymptotic series. In contrast to the problem of resurgent

structures, which has a unique solution, non-perturbative completions are not unique unless we impose additional constraints. For example, under some mild assumptions, one can obtain non-perturbative completions by just considering (lateral) Borel resummations of the asymptotic series. One can enrich this simple completion by adding trans-series. Since general trans-series have arbitrary coefficients, the resurgent analysis of the previous section gives an infinite family of completions. Reality constraints can restrict the values of these coefficients, but it is clear that we need some additional physical input in order to make progress and select a specific non-perturbative completion.

In physical theories, the ultimate arbiter on the correct non-perturbative completion should be comparison to experiment. In topological string theory we don't have such an arbiter (at least for the moment being), and there have been many different proposals for a non-perturbative completion in the literature. Some of these proposals are not fully satisfactory since they do not provide evidence that the would-be non-perturbative functions are actually well-defined. In this section we will consider a non-perturbative proposal which is based on a well-defined quantity, and leads to a rich mathematical structure with many implications. It is motivated by deep physical insights, related to large $N$ dualities and to the quantization of geometry.

### 4.1 Warm-up: the Gopakumar–Vafa duality

Perhaps the simplest non-perturbative completion of a topological string free energy is the GV duality between the resolved conifold and Chern–Simons (CS) theory on the three-sphere [13]. It displays some of the properties of the more general non-perturbative completion that we will introduce in this section, so we will present a brief summary. A more complete treatment can be found in [19, 102].

The inspiration for [13] came from large $N$ dualities between gauge theories and string theories. This is an old idea that goes back to the work of 't Hooft on the $1/N$ expansion and was later implemented in the AdS/CFT correspondence. According to these dualities, a gauge theory with gauge group $U(N)$ and gauge coupling constant $g_s$ is equivalent to a string theory with the same coupling constant. The string theory description emerges in the so-called *'t Hooft limit*, in which $N$ is large but the *'t Hooft parameter* $t = Ng_s$ is kept fixed, i.e.

$$N \to \infty, \qquad g_s \to 0, \qquad t = Ng_s \text{ fixed.} \tag{4.1}$$

In this regime, the observables of the gauge theory have a $1/N$ expansion, i.e. an asymptotic expansion in inverse powers of $1/N$ (note that, since $Ng_s$ is fixed, an expansion in $1/N$ is equivalent to an expansion in $g_s$). For example, in the case of the vacuum free energy of the gauge theory, we have

$$F(g_s, N) \sim \sum_{g \geq 0} F_g(t) g_s^{2g-2}. \tag{4.2}$$

The quantities $F_g(t)$ are then conjectured to be genus $g$ free energies in a dual string theory, and the 't Hooft parameter $t$ corresponds to a geometric modulus in string theory. If this is indeed the case, then the gauge theory quantity $F(g_s, N)$ provides a non-perturbative definition of the total free energy of the string theory. In contrast to what happens in the matrix models of non-critical strings reviewed in C. Johnson's lectures, in large $N$ dualities, like the AdS/CFT correspondence and the Gopakumar–Vafa duality, it is not necessary to take a double-scaling limit, i.e. to tune the 't Hooft parameter to a special value.

Let us also note that, from the point of view of the gauge theory, the modulus is given by a positive integer times the coupling constant, so it is in a sense "quantized" (a similar phenomenon

was already noted in the formula (3.32)). As $N$ becomes large, the discreteness of the 't Hooft parameter should become inessential. More precisely, we expect that in the 't Hooft limit a geometric, continuous description of this parameter will emerge, so that we can identify it with a modulus. This is sometimes interpreted as saying that the continuous or geometric description "emerges" in the large $N$ limit, out of a microscopic description which is not geometric –somewhat similar to the continuous or fluid description of a many-particle system in the thermodynamic limit. We will see below examples of this phenomenon.

In [13], Gopakumar and Vafa found a beautiful realization of a string/gauge theory duality in the realm of topological theories. They considered $U(N)$ Chern–Simons theory, a topological field theory in three dimensions studied and essentially solved by Witten in [103]. Witten found in particular a closed formula for the partition function of this theory on the three-sphere $\mathbb{S}^3$, which reads,

$$Z^{\mathrm{CS}}(g_s, N) = \left(\frac{g_s}{2\pi}\right)^{N/2} \prod_{j=1}^{N-1} \left(2\sin\frac{g_s j}{2}\right)^{N-j}. \tag{4.3}$$

In this expression, $g_s$ is the CS coupling constant, which is related to the so-called CS level $k \in \mathbb{Z}$ by the equation

$$g_s = \frac{2\pi}{k+N}. \tag{4.4}$$

A relatively simple computation done in [13] and reviewed in [19] shows that the free energy $F^{\mathrm{CS}}(g_s, N) = \log Z^{\mathrm{CS}}(g_s, N)$ has an asymptotic expansion of the form (4.2), with

$$F_0^{\mathrm{CS}}(t) = -\frac{t^3}{12} + \frac{\pi \mathrm{i}}{4}t^2 + \frac{\pi^2 t}{6} - \zeta(3) + \mathrm{Li}_3(\mathrm{e}^{-t}),$$

$$F_1^{\mathrm{CS}}(t) = -\frac{t - \pi \mathrm{i}}{24} + \zeta'(-1) + \frac{1}{12}\log g_s + \frac{1}{12}\mathrm{Li}_1(\mathrm{e}^{-t}), \tag{4.5}$$

$$F_g^{\mathrm{CS}}(t) = 2c_g + \frac{(-1)^{g-1}B_{2g}}{2g(2g-2)!}\mathrm{Li}_{3-2g}(\mathrm{e}^{-t}), \qquad g \geq 2.$$

In these equations,

$$t = \mathrm{i}g_s N, \tag{4.6}$$

and $c_g$ is given in (2.8). These are the free energies of the resolved conifold (2.24), up to a polynomial piece in $t$ which can be regarded as the perturbative part of the free energy. Therefore, at least at the level of free energies, there is a large $N$ duality between topological strings on the resolved conifold, and $U(N)$ Chern–Simons gauge theory on the three-sphere.

Let us note that the exact CS free energy is a function of a positive integer $N$ and the coupling $g_s$. The expression (4.3) is manifestly well-defined, as required by a non-perturbative approach. It makes sense in principle for any complex value of $g_s$, although the free energy is singular for some special values of $g_s$. In the topological string side, the 't Hooft parameter is identified with a complexified Kähler parameter and it can be any complex number, as long as $\mathrm{Re}(t) > 0$ (in fact, one can consider more general values of $t$ by a so-called flop transition to a different CY phase). This is an example of how a 't Hooft parameter in a gauge theory becomes a geometric modulus. The non-perturbative completion provided by the gauge theory is in principle restricted to values of $g_s$ of the form (4.4), and values of $t$ for which $t/g_s = \mathrm{i}N$.

It turns out that the partition function (4.3) can be written as a matrix integral [102, 104],

$$Z^{\mathrm{CS}}(g_s, N) = \frac{\mathrm{e}^{-\frac{\hbar}{12}N(N^2-1)}}{N!} \int \prod_{i=1}^{N} \frac{\mathrm{d}u_i}{2\pi} \, \mathrm{e}^{-\frac{1}{2\hbar}\sum_{i=1}^{N} u_i^2} \prod_{i<j}\left(2\sinh\frac{u_i - u_j}{2}\right)^2, \tag{4.7}$$

where $\hbar = \mathrm{i}g_s$. This can be regarded as a deformation of the Gaussian matrix model, in which the standard Vandermonde interaction between eigenvalues gets deformed to a sinh interaction. This type of matrix models appears naturally in the localization of three-dimensional supersymmetric field theories [105] (see [106] for a review).

The Gopakumar–Vafa duality has a natural generalization by performing a quotient of both sides by an ADE discrete group $\Gamma$. In the case of $\Gamma = \mathbb{Z}_p$, one finds a duality between CS theory on the lens space $\mathbb{S}^3/\mathbb{Z}_p = L(p, 1)$, and topological string theory on certain CY geometries which engineer $SU(p)$ gauge theories. This duality has been verified in detail in [107] when $p = 2$ by exploiting a matrix model representation of the partition function, similar to the one in (4.7), see [108] for additional verifications and generalizations.

However, in its current form, these Gopakumar–Vafa-like dualities apply only to special toric CY geometries. We will now consider a different duality that holds conjecturally for *any* toric CY: the *topological string/spectral theory (TS/ST) correspondence*. In this correspondence, the dual of the string theory is not a field theory, but a one-dimensional quantum-mechanical model. We will end up however with matrix model representations for the topological string partition function, similar to (4.7).

## 4.2 Quantum mirror curves and non-perturbative partition functions

The TS/ST correspondence was originally triggered by the observation that the genus zero free energy of a local CY manifold looks like a leading order WKB computation for a quantum system whose classical phase space is defined by the mirror curve (2.34). This suggests that the higher genus corrections might be understood as the result of an appropriate quantization of this curve [6].

What does "quantization of the curve" means? Since the natural Liouville form on the phase space (2.34) defined by the curve is $y\mathrm{d}x$, the natural way to to proceed is to promote $x, y$ to canonically conjugate Heisenberg operators $\mathsf{x}, \mathsf{y}$, satisfying the commutation relation

$$[\mathsf{x}, \mathsf{y}] = \mathrm{i}\hbar. \tag{4.8}$$

Mirror curves involve exponentiated variables, so their quantization involves the Weyl operators $\mathrm{e}^{a\mathsf{x}}$, $\mathrm{e}^{b\mathsf{y}}$, obtained by exponentiation of Heisenberg operators. The equation for a mirror curve is a linear combination of terms of the form $\exp(ax + by)$, and as it is well-known the quantization of such a term suffers from ordering ambiguities. These can however be fixed by using Weyl's prescription (see e.g. section 3.2 of [109]),

$$\mathrm{e}^{ax+by} \to \mathrm{e}^{a\mathsf{x}+b\mathsf{y}}. \tag{4.9}$$

We can therefore try to promote the equation of the curve to an operator, but this is in principle ill-defined since the equation is invariant under multiplication by $\exp(\lambda x + \mu y)$, with $\lambda$, $\mu$ arbitrary. To fix this new ambiguity, we have to write the curve in a canonical form, which in the case of mirror curves of genus one is given by

$$P_X(\mathrm{e}^x, \mathrm{e}^y) = \sum_j c_j \mathrm{e}^{a_j x + b_j y} + \kappa = 0, \tag{4.10}$$

where $\kappa$ is the modulus. This is the form that we have used in e.g. (2.45). We now define the operator associated to the toric CY by

$$\mathsf{O}_X = \sum_j c_j \mathrm{e}^{a_j \mathsf{x} + b_j \mathsf{y}}. \tag{4.11}$$

For example, in the case of local $\mathbb{P}^2$, we simply obtain

$$\mathsf{O}_{\mathbb{P}^2} = \mathrm{e}^{\mathsf{x}} + \mathrm{e}^{\mathsf{y}} + \mathrm{e}^{-\mathsf{x}-\mathsf{y}}. \tag{4.12}$$

(A small comment on notation: when the toric CY is the canonical bundle of a complex surface $S$, we will denote $\mathsf{O}_S$ instead of $\mathsf{O}_X$). Another frequently used example is local $\mathbb{F}_0$. The resulting operator in this case is

$$\mathsf{O}_{\mathbb{F}_0} = \mathrm{e}^{\mathsf{x}} + \xi\mathrm{e}^{-\mathsf{x}} + \mathrm{e}^{\mathsf{y}} + \mathrm{e}^{-\mathsf{y}}, \tag{4.13}$$

where $\xi$ is a mass parameter of the CY, and is in principle complex. Let us point out that, in the case of mirror curves of genus $g_\Sigma$, there are $g_\Sigma$ different operators associated to $g_\Sigma$ different canonical forms of the curve, see [15] for a detailed explanation.

What kind of operators are the $\mathsf{O}_X$? First of all, since the momentum operator is exponentiated, it acts as a translation operator on wavefunctions,

$$\mathrm{e}^{a\mathsf{y}}\psi(x) = \psi(x - a\mathrm{i}\hbar). \tag{4.14}$$

Therefore, the operators $\mathsf{O}_X$ are functional-difference operators. Next, we will regard these operators as acting on the Hilbert space $\mathcal{H} = L^2(\mathbb{R})$. Their domain consists of functions in $\mathcal{H}$ which can be extended to a strip $\mathbb{R} \times \mathcal{I}$ in the complex plane, where $\mathcal{I}$ is an interval depending on the operator. These functions should be square integrable along the lines $\mathbb{R} + \mathrm{i}y$, $y \in \mathcal{I}$.

Once we have defined these operators on a Hilbert space, we can ask all the usual questions that we ask in spectral theory or in quantum mechanics. Are they actually self-adjoint? If yes, what are their spectral properties? It turns out that, generically, the operators obtained from mirror curves with $g_\Sigma \geq 1$ are self-adjoint and have a *discrete* spectrum. The discreteness of the spectrum was first shown numerically in some examples [110], and then it was proved in [12, 111, 112] as a consequence of a stronger result. Namely, it was shown that, for a large family of toric CYs $X$, the inverse operator

$$\rho_X = \mathsf{O}_X^{-1} \tag{4.15}$$

exists and is of trace class, i.e. it satisfies

$$\mathrm{Tr}\, \rho_X < \infty. \tag{4.16}$$

This implies that the spectrum of $\rho_X$ is discrete, with an accumulation point at the origin. It follows that the spectrum of $\mathsf{O}_X$ is also discrete. The trace class property is expected to hold for all quantum mirror curves, provided $g_\Sigma \geq 1$, $\hbar > 0$, and some additional positivity conditions are satisfied by the mass parameters of the geometries (for example, in the case of the operator (4.13), the condition on the mass parameter is that $\xi > 0$.)

**Exercise 4.1.** Calculate numerically the spectrum of (4.12) for various values of $\hbar$, by using e.g. the Rayleigh–Ritz method. Useful details can be found in [110]. Show in particular that, for $\hbar = 2\pi$, if we write the spectrum as $\mathrm{e}^{E_n}$, $n = 0, 1, \cdots$, one finds, for the very first levels, the values in table 1. $\qquad\square$

**Remark 4.2.** Although we also use the expression "quantum curve," our approach is very different from the one used by practitioners of topological recursion and explained in the courses by V. Bouchard [27] and K. Iwaki. In that approach, one has a formal (wave)function, written as a perturbative series in $\hbar$, and looks for a formal differential operator which annihilates it. In particular, there is no notion of Hilbert space. In our approach, in contrast, operators acting on $L^2(\mathbb{R})$ are given from the very beginning by Weyl quantization of the mirror curve. Their spectrum and eigenfunctions are well-defined, and one then looks for the relation between these non-perturbative data and the geometric content of the topological string.

| $n$ | $E_n$ |
|---|---|
| 0 | 2.56264206862381937 |
| 1 | 3.91821318829983977 |
| 2 | 4.91178982376733606 |
| 3 | 5.73573703542155946 |
| 4 | 6.45535922844299896 |

**Table 1**: Numerical spectrum of the operator (4.12) for $n = 0, 1, \cdots, 4$, and $\hbar = 2\pi$.

Trace class operators are in many ways the best possible operators in spectral theory. One can show [113] that, if an operator $\rho_X$ is trace class, the traces of $\rho_X^n$ are all finite, for $n \in \mathbb{Z}_{>0}$, and in addition the *Fredholm or spectral determinant*

$$\Xi_X = \det\left(1 + \kappa \rho_X\right) \tag{4.17}$$

is an *entire* function of $\kappa$. The Fredholm determinant can be regarded as an infinite-dimensional generalization of the characteristic polynomial of a Hermitian matrix (see [114] for a nice introduction). Just as the zeroes of the characteristic polynomial give the eigenvalues of the matrix, the spectrum of the operator $\rho_X$ can be read from the zeroes of the Fredholm determinant: if we denote by $\mathrm{e}^{-E_n}$ the eigenvalues of $\rho_X$, $n \in \mathbb{Z}_{\geq 0}$, the Fredholm determinant vanishes at

$$\kappa = -\mathrm{e}^{E_n}. \tag{4.18}$$

In addition, Fredholm determinants of trace class operators satisfy an infinite-dimensional version of the factorization property of a characteristic polynomial: (4.17) has the infinite product representation [113]

$$\Xi_X(\kappa, \hbar) = \prod_{n=0}^{\infty} \left(1 + \kappa \mathrm{e}^{-E_n}\right). \tag{4.19}$$

Note that we can identify $\kappa$ with the modulus of the CY appearing in (4.10).

Since the Fredholm determinant is an entire function, it has a convergent power series expansion around $\kappa = 0$, of the form

$$\Xi_X(\kappa, \hbar) = 1 + \sum_{N=1}^{\infty} Z_X(N, \hbar)\kappa^N. \tag{4.20}$$

We will refer to the coefficients $Z_X(N, \hbar)$ as *fermionic spectral traces*. They can be defined as [113]

$$Z_X(N, \hbar) = \mathrm{Tr}\left(\Lambda^N(\rho_X)\right), \qquad N = 1, 2, \cdots \tag{4.21}$$

In this expression, the operator $\Lambda^N(\rho_X)$ is defined by $\rho_X^{\otimes N}$ acting on $\Lambda^N\left(L^2(\mathbb{R})\right)$. They can be also obtained from the more conventional, "bosonic" traces $\mathrm{Tr}\rho_X^\ell$, since one has

$$\log \Xi_X(\kappa, \hbar) = -\sum_{\ell=1}^{\infty} \frac{(-\kappa)^\ell}{\ell} \mathrm{Tr}\rho_X^\ell. \tag{4.22}$$

**Exercise 4.3.** Let us consider the following symmetric operator $\rho$ acting on the functions $f \in L^2([0,1])$ such that $f(0) = f(1) = 0$. It is defined by its integral kernel

$$\rho(x, y) = \begin{cases} x(1-y), & \text{if } x \leq y, \\ y(1-x), & \text{if } y \leq x, \end{cases} \tag{4.23}$$

where we recall that $\rho(x, y) = \langle x|\rho|y\rangle$. Verify that this operator has the spectrum $\lambda_n = (\pi n)^{-2}$, with eigenfunctions $e_n(x) = \sqrt{2}\sin(n\pi x)$, $n \in \mathbb{Z}_{>0}$. Show that the corresponding Fredholm determinant is given by [114]

$$\Xi(\kappa) = \prod_{n=1}^{\infty} \left(1 + \frac{\kappa}{\pi^2 n^2}\right) = \frac{\sinh\left(\kappa^{1/2}\right)}{\kappa^{1/2}}. \tag{4.24}$$

From this formula and the spectrum, deduce the following expressions for the fermionic and bosonic spectral traces:

$$Z(N) = \frac{1}{(2N+1)!}, \qquad \operatorname{Tr}\rho^\ell = \frac{1}{\pi^{2\ell}}\zeta(2\ell), \tag{4.25}$$

where $\zeta(z)$ is Riemann's zeta function. $\qquad\qquad\square$

Fredholm determinants and fermionic spectral traces play a very important rôle in the TS/ST correspondence. They were identified in [11, 14, 15] as the natural *global* analytic quantities related to the topological string partition function. In particular, it was conjectured in those papers that the fermionic spectral traces $Z_X(N, \hbar)$ are non-perturbative completions of the total topological string free energy in the conifold frame. More precisely, we have the following

**Conjecture 4.4.** Let us consider the 't Hooft limit

$$N \to \infty, \quad \hbar \to \infty, \qquad \frac{N}{\hbar} \qquad \text{fixed.} \tag{4.26}$$

Then, $Z_X(N, \hbar)$ has an asymptotic expansion of the form

$$\log Z_X(N, \hbar) \sim \sum_{g\geq 0} F_g^c(\lambda)g_s^{2g-2}. \tag{4.27}$$

The relation between the parameters in the two sides the following. The string coupling constant is related to $\hbar$ by

$$g_s = \frac{4\pi^2}{\hbar}. \tag{4.28}$$

$F_g^c(\lambda)$ is the genus $g$ free energy of $X$ in the conifold frame, and the 't Hooft parameter

$$\lambda = Ng_s \tag{4.29}$$

is identified with the canonically normalized conifold coordinate.

Let us make various comments on this conjecture.

1. As we emphasized in the Introduction, a *bona fide* non-perturbative definition has to involve a well-defined function, whose asymptotics reproduces the perturbative series that we started with. The fermionic spectral trace is well-defined for any $N \in \mathbb{Z}_{>0}$ and $\hbar > 0$ due to the crucial trace class property of the operator $\rho_X$, which as we mentioned before has been rigorously proved for many toric CYs $X$. It is far from obvious that the asymptotic expansion of this trace gives the perturbative topological string free energies, but this is what makes this non-perturbative definition interesting, and the conjecture 4.4 challenging.

2. The limit (4.26) is indeed very similar to the 't Hooft limit (4.1). An interesting point is that, as shown in (4.28), the coupling constant is essentially the inverse of $\hbar$. Therefore, the weakly coupled limit of the topological string corresponds to the strong coupling limit of the quantum mechanical problem. When the quantization of mirror curves was initially proposed in [6], it was hoped that the topological string would emerge in the weak coupling limit, but it was later realized that this limit corresponds to the NS topological string mentioned in section 3.3. The emergence of the conventional topological string in the strong coupling limit was first observed in a different line of work on localization and ABJM theory [39, 115, 116], see [117, 118] for reviews of these developments.

3. Although the conjecture is formulated in the strong coupling limit of the spectral problem, it turns out that the operators in exponentiated variables appearing in the TS/ST correspondence conjecturally satisfy a strong-weak coupling duality in $\hbar$ [119]: the strong coupling limit of the spectrum $\hbar \to \infty$ can be related to the weak coupling limit $\hbar \to 0$. This is expected to be related to the modular duality for Weyl operators noted by Faddeev in [58].

4. The conjecture 4.4 suggests that $Z_X(N, \hbar)$ plays the rôle of a partition function in a quantum field theory, where $N$ should be interpreted as the rank of a gauge group. Although there is no explicit realization of such a quantum field theory for the moment being, in concrete examples one can relate the fermionic spectral traces to matrix integrals. Indeed, a theorem of Fredholm asserts that, if $\rho_X(p_i, p_j)$ is the kernel of $\rho_X$, the fermionic spectral trace can be computed as an $N$-dimensional integral,

$$Z_X(N, \hbar) = \frac{1}{N!} \int \det\left(\rho_X(p_i, p_j)\right) \, \mathrm{d}^N p. \tag{4.30}$$

In cases where the kernel $\rho_X$ can be computed explicitly, the above integral can be written as an eigenvalue integral, and analyzed with matrix model techniques [11, 12, 120], as we will see in examples in the next section.

5. Physically, (4.30) can be interpreted as the canonical partition function of a non-interacting Fermi gas described by the one-body density matrix $\rho_X$. It is then natural to define the Hamiltonian $\mathsf{H}_X$ of such a system by $\rho_X = \mathrm{e}^{-\mathsf{H}_X}$. In this picture, the Fredholm determinant $\Xi_X(\kappa)$ is the grand-canonical partition function of the Fermi gas, and $\kappa = \mathrm{e}^\mu$ is the exponent of the fugacity in the grand-canonical ensemble.

6. The conjecture 4.4 is in fact a consequence of a stronger conjecture formulated in [14]. This conjecture gives an *exact* expression for $Z_X(N, \hbar)$ and for the Fredholm determinant $\Xi_X(\kappa)$ in terms of the GV free energy and additional enumerative information, see e.g. [40] for a review.

## 4.3 Local $\mathbb{P}^2$, non-perturbatively

The conjecture 4.4 seems difficult to prove, since it relates a quantum mechanical model at strong coupling (i.e. for $\hbar \to \infty$) to topological string theory on a toric CY theefold. It is however a falsifiable statement, i.e. we can calculate both sides of the conjecture and check whether they are equal or not. So far all tests have been successful. Many of these tests involve the stronger form of the conjecture mentioned above, and they are typically numerical, since it is easier to calculate the fermionic spectral traces numerically for low values of $N$, than to compute their

asymptotic behavior in the 't Hooft limit. In some cases, however, it is also possible to calculate this asymptotic expansion analytically. We will now consider the spectral theory associated to local $\mathbb{P}^2$, where many concrete results can be obtained.

The starting point of this analysis is the fact that, for some mirror curves, the integral kernel of the operator $\rho_X$ can be explicitly computed. This is remarkable since there are not many trace class operators in conventional one-dimensional quantum mechanics where this can be done.

Let us consider the following family of operators:

$$\mathsf{O}_{m,n} = \mathrm{e}^{\mathsf{x}} + \mathrm{e}^{\mathsf{y}} + \mathrm{e}^{-m\mathsf{x}-n\mathsf{y}}, \quad m, n \in \mathbb{R}_{>0}. \tag{4.31}$$

They were called three-term operators in [111]. Note that the case $m = n = 1$ corresponds to local $\mathbb{P}^2$ (the case with arbitrary positive integers $m, n$ corresponds to the toric CY given by the canonical bundle on the weighted projective space $\mathbb{P}(1, m, n)$). We now define the function

$$\Psi_{a,c}(x) = \frac{\mathrm{e}^{2\pi a x}}{\Phi_{\mathsf{b}}(x - \mathrm{i}(a + c))}, \tag{4.32}$$

involving Faddeev's quantum dilogarithm, where $a, c$ are positive real numbers. Let us now introduce normalized Heisenberg operators $\mathsf{q}, \mathsf{p}$, satisfying the normalized commutation relation

$$[\mathsf{p}, \mathsf{q}] = (2\pi\mathrm{i})^{-1}. \tag{4.33}$$

They are related to $\mathsf{x}, \mathsf{y}$ by the linear canonical transformation,

$$\mathsf{x} = 2\pi\mathsf{b}\frac{(n+1)\mathsf{p} + n\mathsf{q}}{m+n+1}, \quad \mathsf{y} = -2\pi\mathsf{b}\frac{m\mathsf{p} + (m+1)\mathsf{q}}{m+n+1}. \tag{4.34}$$

In particular, $\hbar$ is related to $\mathsf{b}$ by

$$\hbar = \frac{2\pi\mathsf{b}^2}{m+n+1}. \tag{4.35}$$

It was proved in [111] that, in the momentum representation associated to $\mathsf{p}$, the integral kernel of the operator $\rho_{m,n}$ can be written explicitly in terms of the function (4.32). It reads,

$$\rho_{m,n}(p, p') = \frac{\overline{\Psi_{a,c}(p)}\,\Psi_{a,c}(p')}{2\mathsf{b}\cosh\left(\pi\frac{p-p'+\mathrm{i}(a+c-nc)}{\mathsf{b}}\right)}. \tag{4.36}$$

In this equation, $a$, $c$ are given by

$$a = \frac{m\mathsf{b}}{2(m+n+1)}, \qquad c = \frac{\mathsf{b}}{2(m+n+1)}. \tag{4.37}$$

As we will see in the following exercises, this results allows for many explicit calculations.

**Exercise 4.5.** In this exercise you are asked to use the explicit formula (4.36) to calculate the first trace of $\rho_{m,n}$, given by

$$\mathrm{Tr}\rho_{m,n} = \int_{\mathbb{R}} \rho_{m,n}(p, p)\mathrm{d}p. \tag{4.38}$$

First note that, by using the property (B.8), one can write

$$|\Psi_{a,c}(p)|^2 = \mathrm{e}^{4\pi a p}\frac{\Phi_{\mathsf{b}}(p + \mathrm{i}(a + c))}{\Phi_{\mathsf{b}}(p - \mathrm{i}(a + c))}, \tag{4.39}$$

therefore

$$\rho_{m,n}(p,p) = \frac{\Phi_{\mathsf{b}}(p + \mathsf{i}(a+c))}{\Phi_{\mathsf{b}}(p - \mathsf{i}(a+c))} \frac{e^{4\pi ap}}{2\mathsf{b}\cos\left(\pi\frac{a+c-nc}{\mathsf{b}}\right)}. \tag{4.40}$$

The first trace can be computed explicitly for arbitrary values of $m, n$ and $\hbar$, by using properties of Faddeev's quantum dilogarithm, as shown in [111]. For this exercise we will consider the case

$$\hbar = 2\pi. \tag{4.41}$$

This is a special value of $\hbar$ where the theory of quantum mirror curves simplifies very much, as shown in [14]. We will also take $n = 1$ and $m$ an arbitrary positive integer. For these values we have

$$\mathsf{b}^2 = m + 2, \tag{4.42}$$

and

$$a + c = \frac{1}{2}\left(\mathsf{b} - \mathsf{b}^{-1}\right). \tag{4.43}$$

Show, by using the properties (B.9a), (B.9b), that

$$\frac{\Phi_{\mathsf{b}}\left(p + \frac{\mathsf{i}}{2}\left(\mathsf{b} - \mathsf{b}^{-1}\right)\right)}{\Phi_{\mathsf{b}}\left(p - \frac{\mathsf{i}}{2}\left(\mathsf{b} - \mathsf{b}^{-1}\right)\right)} = \frac{1 - e^{2\pi\mathsf{b}^{-1}p}}{1 - e^{2\pi\mathsf{b}p}}. \tag{4.44}$$

Deduce the following expression:

$$\mathrm{Tr}\rho_{m,1}\left(\hbar = 2\pi\right) = \frac{1}{2\pi\cos\left(\frac{\pi m}{2(m+2)}\right)}\int_{\mathbb{R}} e^{(m-1)y}\frac{\sinh(y)}{\sinh((m+2)y)}\mathrm{d}y, \tag{4.45}$$

where $y = \pi p/\mathsf{b}$. The integral can be evaluated e.g. by residues, and one concludes that [111]

$$\mathrm{Tr}\rho_{m,1}\left(\hbar = 2\pi\right) = \frac{1}{4(m+2)\sin\left(\frac{\pi}{m+2}\right)\sin\left(\frac{2\pi}{m+2}\right)}. \tag{4.46}$$

In particular, for $m = 1$, which corresponds to local $\mathbb{P}^2$, one finds

$$\mathrm{Tr}\rho_{1,1}\left(\hbar = 2\pi\right) = \frac{1}{9}. \tag{4.47}$$

$\square$

**Exercise 4.6.** C. Johnson explained in his lectures how to compute Fredholm determinants numerically with the approach of [121], in the context of matrix models of 2d gravity (see [122, 123]). This approach requires an explicit knowledge of the integral kernel, which for three-term operators is given by (4.36). In the previous exercise we showed that the diagonal kernel $\rho_{m,1}(p,p)$ simplifies for $\hbar = 2\pi$. We can in fact simplify the whole integral kernel by using a similarity transformation [124],

$$\rho_X(p,p') \to h(p)\rho_X(p,p')(h(p'))^{-1}, \tag{4.48}$$

where $h(p)$ is non-vanishing function. Such a transformation does not change the value of the traces of $\rho_X^n$, nor the spectral determinant. In the case of $\rho_{m,n}(p,p')$, we take

$$h(p) = \sqrt{\frac{\Psi_{a,c}(p)}{\Psi_{a,c}(p)}}. \tag{4.49}$$

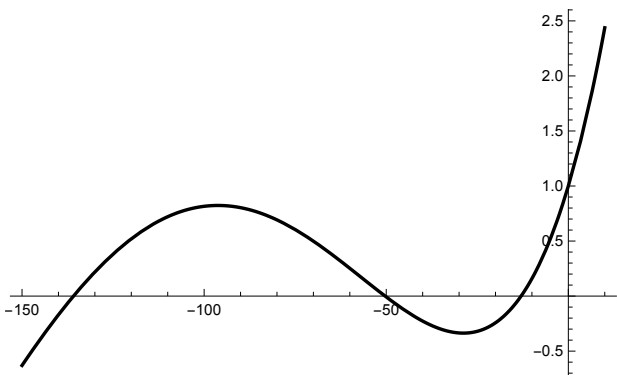

**Figure 8**: A numerical calculation of the Fredholm determinant $\Xi_{\mathbb{P}^2}(\kappa)$ for local $\mathbb{P}^2$, for $\hbar = 2\pi$.

Show that, for $\hbar = 2\pi$, and after the similarity transformation, we can write

$$\rho_{1,1}(y, y') = \frac{1}{2\pi} \sqrt{\frac{\sinh(y)}{\sinh(3y)}} \frac{1}{\cosh\left(y - y' + \frac{i\pi}{6}\right)} \sqrt{\frac{\sinh(y')}{\sinh(3y')}} \tag{4.50}$$

in terms of the variable $y$ appearing in (4.45). Implement the algorithm of [121] to calculate $\Xi_{\mathbb{P}^2}(\kappa)$ numerically, and use this result to obtain the very first energy levels. As an example of what you should find, in Fig. 8 I plot a numerical calculation of $\Xi_{\mathbb{P}^2}(\kappa)$, obtained as follows. First, I use a Gauss–Kronrod quadrature or order 50 to calculate the Fredholm determinant. I do this calculation for 150 values of $\kappa$ in the interval $[-150, 10]$, and I construct an interpolating function. The zeroes of this interpolating function are approximately at $-12.97004$, $-50.3105$ and $-135.882$ (rounded numerical values), in good agreement with the results in table 1 (remember the relation (4.18)). Note that the plot in Fig. 8 is indistinguishable from the plot of the same quantity that appears in [40]. The latter was obtained by using the conjecture of [14], which expresses the Fredholm determinant as a "quantum theta function." The fact that they agree so precisely (within numerical errors) is an explicit test of the TS/ST correspondence. $\qquad\square$

By using (4.36) one can write down an explicit expression for the integral (4.30), which we will denote in this case as $Z_{m,n}(N, \hbar)$. As in [116, 125], we use Cauchy's identity:

$$\frac{\prod_{i<j}\left[2\sinh\left(\frac{\mu_i - \mu_j}{2}\right)\right]\left[2\sinh\left(\frac{\nu_i - \nu_j}{2}\right)\right]}{\prod_{i,j} 2\cosh\left(\frac{\mu_i - \nu_j}{2}\right)} = \det_{ij} \frac{1}{2\cosh\left(\frac{\mu_i - \nu_j}{2}\right)}$$

$$= \sum_{\sigma \in S_N} (-1)^{\epsilon(\sigma)} \prod_i \frac{1}{2\cosh\left(\frac{\mu_i - \nu_{\sigma(i)}}{2}\right)}. \tag{4.51}$$

One finds then,

$$Z_{m,n}(N, \hbar) = \frac{1}{N!} \int_{\mathbb{R}^N} \frac{\mathrm{d}^N p}{\mathsf{b}^N} \prod_{i=1}^N |\Psi_{a,c}(p_i)|^2 \frac{\prod_{i<j} 4\sinh\left(\frac{\pi}{\mathsf{b}}(p_i - p_j)\right)^2}{\prod_{i,j} 2\cosh\left(\frac{\pi}{\mathsf{b}}(p_i - p_j) + i\pi C_{m,n}\right)}, \tag{4.52}$$

where

$$C_{m,n} = \frac{m - n + 1}{2(m + n + 1)}. \tag{4.53}$$

The matrix integral (4.52) is real and convergent for $\hbar > 0$, since the kernel (4.36) is Hermitian and trace class. This is in contrast to doubly-scaled matrix models of two-dimensional gravity, which are often ill-defined non-perturbatively, at least with the standard choice of integration contours.

In order to test the conjecture 4.4 we have to study the matrix integral (4.52) in the 't Hooft limit (4.26), therefore we should understand what happens to the integrand of (4.52) when $\hbar$ (or equivalently $\mathsf{b}$) is large. To do this, we first change variables to

$$u_i = \frac{2\pi}{\mathsf{b}} p_i, \tag{4.54}$$

and we introduce the parameter

$$\mathsf{g} = \frac{1}{\hbar} = \frac{m+n+1}{2\pi} \frac{1}{\mathsf{b}^2}, \tag{4.55}$$

so that the weak coupling regime of $\mathsf{g}$ is the strong coupling regime of $\hbar$. In general quantum-mechanical models, this regime is difficult to understand, but in this case we can use the crucial property of self-duality of Faddeev's quantum dilogarithm,

$$\Phi_\mathsf{b}(x) = \Phi_{1/\mathsf{b}}(x). \tag{4.56}$$

Then, by using (4.39), we can write

$$|\Psi_{a,c}(p)|^2 = \exp\left(\frac{mu}{2\pi\mathsf{g}}\right) \frac{\Phi_{1/\mathsf{b}}\left((u + 2\pi\mathsf{i}(a+c)/\mathsf{b})/2\pi\mathsf{b}^{-1}\right)}{\Phi_{1/\mathsf{b}}\left((u - 2\pi\mathsf{i}(a+c)/\mathsf{b})/2\pi\mathsf{b}^{-1}\right)}, \tag{4.57}$$

where $u$ and $p$ are related through (4.54). When $\mathsf{b}$ is large, $1/\mathsf{b}$ is small and we can use the asymptotic expansion (B.10). We define the *potential* of the matrix model as,

$$V_{m,n}(u, \mathsf{g}) = -\mathsf{g}\log|\Psi_{a,c}(p)|^2, \tag{4.58}$$

where $u$ and $p$ are related as in (4.54). By using (B.10), we deduce that this potential has an asymptotic expansion at small $\mathsf{g}$, of the form

$$V_{m,n}(u, \mathsf{g}) = \sum_{\ell \geq 0} \mathsf{g}^{2\ell} V_{m,n}^{(\ell)}(u). \tag{4.59}$$

The leading contribution as $\mathsf{g} \to 0$ is given by the "classical" potential,

$$V_{m,n}^{(0)}(u) = -\frac{m}{2\pi} u - \frac{m+n+1}{2\pi^2} \mathrm{Im}\left(\mathrm{Li}_2\left(-\mathrm{e}^{u+\pi\mathsf{i}\frac{m+1}{m+n+1}}\right)\right). \tag{4.60}$$

**Exercise 4.7.** By using the asymptotics of the dilogarithm,

$$\mathrm{Li}_2(-\mathrm{e}^x) \approx \begin{cases} -x^2/2, & x \to \infty, \\ -\mathrm{e}^x, & x \to -\infty, \end{cases} \tag{4.61}$$

show that

$$V_{m,n}^{(0)}(u) \approx \begin{cases} \frac{u}{2\pi}, & u \to \infty, \\ -\frac{m}{2\pi} u, & u \to -\infty, \end{cases} \tag{4.62}$$

i.e. it is a linearly confining potential. This is similar to the potentials appearing in matrix models for Chern–Simons–matter theories (see e.g. [116]). □

We can now write the matrix integral (4.52) as

$$Z_{m,n}(N,\hbar) = \frac{1}{N!} \int_{\mathbb{R}^N} \frac{\mathrm{d}^N u}{(2\pi)^N} \prod_{i=1}^{N} \mathrm{e}^{-\frac{1}{\mathsf{g}} V_{m,n}(u_i,\mathsf{g})} \frac{\prod_{i<j} 4 \sinh\left(\frac{u_i-u_j}{2}\right)^2}{\prod_{i,j} 2 \cosh\left(\frac{u_i-u_j}{2} + \mathrm{i}\pi C_{m,n}\right)}. \tag{4.63}$$

This expression is very similar to matrix models that have been studied before in the literature. The interaction between eigenvalues is similar to the matrix model (4.7) which appears in the Gopakumar–Vafa duality, and is identical to the one appearing in the generalized $O(2)$ models of [126], and in some matrix models for Chern–Simons–matter theories studied in for example [125]. The parameter $\mathsf{g}$ corresponds to the string coupling constant, but in contrast to conventional matrix models, the potential depends itself on $\mathsf{g}$.

The above expression is perfectly suited to study the 't Hooft limit. One can use e.g. saddle-point techniques to solve for the planar limit, i.e. for the leading behavior in the $1/N$ expansion. This limit is described by the so-called planar resolvent and density of eigenvalues, and from this one can compute the genus zero free energy $F_0^c(\lambda)$. In this limit, only the classical part of the potential (4.60) has to be taken into account. The resolvent for the local $\mathbb{P}^2$ matrix model, given by (4.63) with $m = n = 1$, was first conjectured in [127]. This conjecture was later proved in a *tour de force* calculation in [120], by extending the techniques of [126]. The result for the resolvent and density of eigenvalues is as follows. Let us introduce the exponentiated variable

$$X = \mathrm{e}^u, \tag{4.64}$$

where $u$ is the variable appearing in (4.63). Then, the spectral curve describing the planar limit of the matrix model turns out to be given by

$$X^3 + Y^{-3} + \kappa X Y^{-1} + 1 = 0, \tag{4.65}$$

which can be easily seen to be a reparametrization of the classical mirror curve (2.45). As it is standard in the study of the planar limit, the density of eigenvalues can be written as the discontinuity of the so-called planar resolvent [128] (see [102, 129] for a review),

$$\rho(u) = \frac{1}{2\pi \mathrm{i}} \left( \omega(X - \mathrm{i}0) - \omega(X + \mathrm{i}0) \right), \tag{4.66}$$

where in this case

$$\omega(X) = \frac{3\mathrm{i}}{\pi} \frac{\log Y(X)}{X}. \tag{4.67}$$

(The actual planar resolvent has an additional piece, but it does not contribute to the density of eigenvalues). The density $\rho(u)$ is a symmetric one-cut distribution, and the end-points of the cut can be found from the branch cuts of the spectral curve. They are given by $\pm a$, where

$$a = -\frac{1}{3} \log \left( -\frac{2}{27} \kappa^3 - 1 - \frac{2}{27} \sqrt{\kappa^6 + 27 \kappa^3} \right), \tag{4.68}$$

and we are assuming that $-\infty < \kappa < -3$ which corresponds to the region $-1/27 < z < 0$. One way of testing the result above for $\rho(u)$ is to consider a finite but large number $N$ of "typical eigenvalues" and see how they distribute along a histogram. This distribution should approximate the density of eigenvalues $\rho(u)$ as $N$ grows large. In the lectures by C. Johnson such a distribution was obtained in the case of the Gaussian matrix model by generating a random matrix with

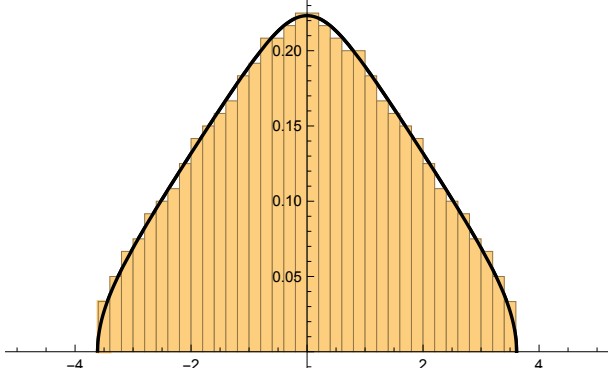

**Figure 9**: This figure shows a histogram of the equilibrium eigenvalues of the local $\mathbb{P}^2$ matrix model, for $N = 600$ and $\kappa = -70$, together with the density of eigenvalues $\rho(u)$ (the black line).

Gaussian weight, and then calculating their eigenvalues. Here the probability distribution is not Gaussian, and we will obtain the distribution in a different way, as follows. Let us consider an eigenvalue integral of the form

$$Z(N) = \int \prod_{i=1}^{N} e^{-\sum_{i=1}^{N} v(x_i)} \prod_{i<j} \mathcal{I}(x_i - x_j). \tag{4.69}$$

The saddle point at finite $N$ is the configuration $x_1, \cdots, x_N$ that minimizes the "effective action"

$$S(x_1, \cdots, x_N) = \sum_{i=1}^{N} v(x_i) - \sum_{i<j} \log \mathcal{I}(x_i - x_j). \tag{4.70}$$

This action can be regarded as a generalization of the Dyson gas. In the limit of large $N$ the minimization is described by the eigenvalue distribution $\rho(x)$, but the minimization problem can be solved for any finite $N$, under appropriate conditions. We will call the $x_1, \cdots, x_N$ minimizing (4.70) the *equilibrium eigenvalues*[1]. To find these eigenvalues in the case of (4.63) we use only the classical potential (4.60). In Fig. 9 we show both, the histogram for the equilibrium eigenvalues obtained from the matrix model for local $\mathbb{P}^2$ with $N = 600$ and $\kappa = -70$, and the density of eigenvalues $\rho(u)$ obtained from (4.66), (4.67). The latter provides an excellent approximation to the former.

The calculation of the subleading terms of the $1/N$ expansion is complicated. Ideally, one would like to show that the matrix integral (4.63) satisfies the topological recursion of [48], and together with the remodeling approach of [10], one would have a proof of the conjecture 4.4 for local $\mathbb{P}^2$ and some other cases. This has not been achieved so far. However, it is still possible to test the conjecture 4.4 by calculating the matrix integral in a perturbative expansion in **g**, at fixed $N$. This can be used to obtain the expansion of $F_g^c(\lambda)$ around $\lambda = 0$, as shown in detail [11], and it can be verified that the result agrees with (2.62), (2.63).

---

[1]This procedure is different from the one discussed in C. Johnson's lectures, see also [122]. He looks at realizations of a random Gaussian distribution at finite $N$, and different realizations lead to different lists of eigenvalues. Since the large $N$ limit implements at the same time a classical and a thermodynamic limit, his calculation takes into account both quantum and finite size effects at finite $N$. In my calculation I consider already the classical limit of the problem, described by the generalized Dyson gas (4.70), and therefore by working at finite $N$ I take into account only finite size effects. Of course, as $N$ becomes large, both calculations converge to the same probability distribution.

Let us close this section with some additional comments on the TS/ST correspondence.

1. The discussion above makes it clear that the conjecture 4.4 can be regarded as an explicit realization of "quantum geometry," in which the spacetime geometry of the toric CY, together with its "stringy" corrections (given by embedded Rieman surfaces) *emerges* from a simple quantum mechanical model on the real line. As in other string/gauge theory dualities of the 't Hooft type, in the quantum model the CY modulus $\lambda$ is "quantized" in units of the string coupling constant. We can also think about the geometry of the CY as emerging from the eigenvalue integral (4.30) in the 't Hooft limit. This limit is encoded in a spectral curve, which is nothing but the mirror curve we started with. Here we have seen how this works in the explicit example of local $\mathbb{P}^2$, and one can also work out the case of local $\mathbb{F}_0$ [120, 127]. We expect however this picture to hold for general toric manifolds, namely the 't Hooft limit of the spectral traces should be described by a spectral curve given by the mirror curve. Higher order corrections to the spectral trace in the $1/N$ expansion should be governed by the topological recursion.

2. We have provided a non-perturbative definition of topological string theory on toric CY manifolds in terms of simple quantum-mechanical models. The reader could ask why is this definition special. Non-perturbative definitions are not unique, and the TS/ST correspondence has not been justified so far as a full-fledged string/gauge theory duality. One can argue however that this non-perturbative definition is physically and mathematically very rich. When applied to local $\mathbb{F}_0$ it gives an exact description of the partition function of ABJM theory on the three-sphere as a sum over worldsheet and membrane contributions in type IIA/M-theory (see e.g. [117, 118] for reviews and references). It leads to explicit conjectures on the exact spectrum of quantum mirror curves, and by using the formulation of [119], it also leads to exact quantization conditions [130, 131] for cluster integrable systems associated to toric CY threefolds [8]. It makes surprising predictions on the classical problem of evaluating CY periods at the conifold point, which have been verified in many cases with sophisticated tools in algebraic geometry [132–134].

3. It was shown in [135–137] that there is a very interesting "dual" 4d limit of the conjecture 4.4 which makes contact with $\mathcal{N} = 2$ gauge theories in four dimensions. The simplest example is SW theory, which has been known for a long time to be engineered by the local $\mathbb{F}_0$ geometry [4]. In the dual 4d limit, the fermionic spectral trace for local $\mathbb{F}_0$ appearing in the l.h.s. of (4.27) becomes the matrix integral

$$Z(N; g_s) = \frac{1}{N!} \int \frac{\mathrm{d}^N x}{(4\pi)^N} \prod_{i=1}^{N} e^{-\frac{4}{g_s} \cosh(x_i)} \prod_{i<j} \tanh^2 \left( \frac{x_i - x_j}{2} \right). \qquad (4.71)$$

In the same limit, the genus $g$ free energies appearing in the r.h.s. of (4.27) become the SW free energies in the magnetic frame $\mathcal{F}_g$, discussed in section 3.3. As a particular case of (4.27) one obtains then the conjecture that, in the 't Hooft limit, the matrix integral (4.71) has the asymptotic expansion

$$\log Z(N; g_s) \sim \sum_{g \geq 0} \mathcal{F}_g(t) g_s^{2g-2}, \qquad (4.72)$$

where the 't Hooft parameter $t = N g_s$ is identified with the flat coordinate $-\mathrm{i} a_D(u)$. This last statement can be proved rigorously [135], providing in this way a proof of the conjecture

4.4 in this special case. The proof can be generalized to arbitrary $SU(N)$, $\mathcal{N} = 2$ pure super Yang–Mills theories [137]. This gives further evidence for the TS/ST correspondence.

4. The spectral theory side guarantees that the fermionic trace $Z_X(N, \hbar)$ is well-defined for $N \in \mathbb{Z}_{>0}$, $\hbar > 0$, provided the mass parameters satisfy some positivity constraints. This might be regarded as too restrictive, specially in view of e.g. approaches based on Borel resummation where it is natural to consider complex values of $\hbar$. Before filing a complaint, one should note that this is a natural aspect of dualities between strings and quantum theories, since the latter require various conditions to be well defined, and these include reality and positivity constraints on some of their parameters. However, there are indications that one can analytically continue the fermionic traces to functions with a much larger analyticity domain. One can use e.g. non–Hermitian extensions of quantum mechanics to consider complex values of $\hbar$ and of the mass parameters [138–140], and we expect the fermionic traces to be analytic on the cut complex plane $\mathbb{C}' = \mathbb{C}\backslash(-\infty, 0]$ of the $\hbar$ variable [3, 14, 111].

5. We have focused on relatively complicated examples, in which the mirror curve has genus $g_\Sigma = 1$. What happens to the topological string on the resolved conifold in the context of the TS/ST correspondence, which after all is a simpler example? It turns out that the quantization of the mirror curve to the resolved conifold gives the three-term operator (4.31) with $m = 1$, $n = -1$. This operator is not trace class, so strictly speaking the standard version of the correspondence does not apply in that case. Fortunately, there is a beautiful way to make sense of the spectral theory of the resolved conifold due to Y. Hatsuda [141], in which one considers an appropriate analytic continuation of the spectral zeta function, instead of the fermionic spectral traces. The resulting theory leads e.g. to a spectral determinant in agreement with the general theory proposed in [14, 39].

6. As a final comment, one should ask what is the relation between the contents of this section and the resurgent story explained in section 3. Resurgence suggests that the asymptotic expansion (4.27) can be promoted to an exact formula for the fermionic spectral traces, by using (lateral) Borel resummation and including perhaps trans-series. This formula would be of the form,

$$\log Z_X(N, \hbar) = s_\pm (F^c) (N, \hbar) + \cdots \tag{4.73}$$

where $F^c$ is the perturbative topological string series in the conifold frame, and the dots in the r.h.s. represent possible additional trans-series. Optimistically, these trans-series are the ones appearing in the resurgent structure unveiled in section 3. This issue was addressed in [63] in the case of local $\mathbb{P}^2$, where strong numerical evidence was given that the exact fermionic spectral traces are *different* from the Borel resummation of the perturbative series. For example, when $N = 1$ and $\hbar = 2\pi$, the Borel resummation gives

$$s_\pm (F^c) (1, 2\pi) = -2.197217... \tag{4.74}$$

while the exact result was obtained in (4.47),

$$\log Z_{\mathbb{P}^2}(1, 2\pi) = -\log(9) = -2.197224... \tag{4.75}$$

Therefore, additional trans-series corrections are clearly needed. In [63] some numerical evidence was given that the trans-series appearing in the resurgent structure can provide

the required corrections, but more work should be done in order to understand the non-perturbative effects implicit in the TS/ST correspondence. On a more conceptual level, note that, in contrast to Borel resummations, which jump along Stokes rays in the complex plane of the coupling constant, the fermionic spectral traces are analytic on a subset of the complex plane, conjecturally the cut plane $\mathbb{C}'$ mentioned above.

## 5 Open problems

The approach to non-perturbative topological strings proposed in these lectures leads to the conjecture 3.5 on the resurgent structure of the topological string, and to the conjecture 4.4 on a non-perturbative definition in the toric case, in terms of quantum mechanics. These conjectures lead to many open problems. Some of them have been already mentioned in the lectures, and we will conclude by listing a few more.

1. It would be wonderful to prove these conjectures rigorously, assuming they are true, but this seems to be extremely difficult in both cases.

2. Some aspects of the conjecture 3.5 have been tested, but in most cases the determination of Stokes constants can be only done numerically, and for a limited number of Borel singularities. This makes it difficult to verify that they agree with the BPS invariants of the underlying CY threefold. More work is needed to compute Stokes constants, and it would be of course of paramount importance to develop analytic techniques to obtain them.

3. In some cases, knowledge of the resurgent structure of a perturbative series makes it possible to determine its resummation as the solution to a Riemann–Hilbert problem. This was first suggested by Voros [142], but in the last years, starting with the work of [143], it has been understood that this problem can be formulated in terms of a set of TBA-like integral equations (see e.g. [144]). It would be interesting to pursue these ideas, perhaps by using mould techniques from the theory of resurgence (see e.g. [145]).

4. The TS/ST correspondence has been extensively tested, and some particular cases have been proved rigorously. However, there are still many open problems. For example, we don't know how to find exact integral kernels for generic quantum mirror curves (the simplest case in which this problem has not been solved is the operator associated to local $\mathbb{F}_1$). It would be extremely interesting to find explicit answers for the integral kernels in other cases.

5. The "dual" 4d limit of quantum mirror curves formulated in [135] leads to operators on the real line describing topological strings on SW-like curves. However, we lack an intrinsic description of these operators in terms of the curve itself (so far, these operators can be described in detail only when the integral kernel of the "parent" operator is known exactly, as in [135, 136]). Perhaps these operators are related to the quantization of curves as defined by topological recursion.

6. We also mentioned that the fermionic spectral traces in the TS/ST correspondence resemble partition functions of three-dimensional quantum field theories. It would be interesting to make this concrete, and/or to provide a string theory/QFT perspective on this mysterious correspondence.

## Acknowledgements

I would like to thank the organizers of the Les Houches school on quantum geometry (Bertrand Eynard, Elba García-Failde, Alessandro Giachetto, Paolo Gregori, and Danilo Lewanski) for inviting me to deliver these lectures. Alba Grassi, Jie Gu, Ricardo Schiappa and Maximilian Schwick read preliminary versions of these lectures and made very useful observations. I would also like to thank my collaborators on the topics discussed here. My work has been supported in part by the ERC-SyG project "Recursive and Exact New Quantum Theory" (ReNewQuantum), which received funding from the European Research Council (ERC) under the European Union's Horizon 2020 research and innovation program, grant agreement No. 810573.

## A  A short review of resurgence

In this section we will review some of the results in resurgence which we will need. I will be brief since I. Aniceto has covered the topic in this school. A wonderful introduction to the subject can be found in [146]. More formal developments can be studied in [147, 148]. A more physical perspective can be found in [129, 149, 150].

Let

$$\varphi(z) = \sum_{n \geq 0} a_n z^n \tag{A.1}$$

be a factorially divergent, formal power series in $z$, i.e. $a_n \sim n!$ (such series are also called Gevrey-1). Its *Borel transform* is given by

$$\widetilde{\varphi}(\zeta) = \sum_{n \geq 1} a_n \frac{\zeta^{n-1}}{(n-1)!}. \tag{A.2}$$

**Remark A.1.** The above definition of the Borel transform is the one used e.g. in [129, 148], and in K. Iwaki's lectures. There is another definition used in e.g. [149] and given by

$$\widehat{\varphi}(\zeta) = \sum_{n \geq 0} a_n \frac{\zeta^n}{n!}. \tag{A.3}$$

It is related to the previous definition by

$$\widetilde{\varphi}(\zeta) = \frac{\mathrm{d}\widehat{\varphi}(\zeta)}{\mathrm{d}\zeta}. \tag{A.4}$$

A *resurgent function* is a Gevrey-1 series $\varphi(z)$ whose Borel transform has the following property: on any line issuing from the origin, there are only a finite number of singularities of the Borel transform, and $\widetilde{\varphi}(\zeta)$ may be continued analytically along any path that follows the line, while circumventing (from above or from below) those singular points. This is Ecalle's principle of *endless analytic continuation*. A resurgent function is *simple* if the singularities of its Borel transform are poles or logarithmic branch cuts.

Let us assume that $\widetilde{\varphi}(\zeta)$ is a simple resurgent function with a logarithmic singularity at $\zeta = \zeta_\omega$. Its local expansion there is of the form

$$\widetilde{\varphi}(\zeta_\omega + \xi) = -\frac{\mathsf{S}}{2\pi} \log(\xi) \sum_{n \geq 0} \tilde{c}_n \xi^n + \text{regular}, \tag{A.5}$$

where the series

$$\sum_{n\geq 0}\tilde{c}_n\xi^n \tag{A.6}$$

has a finite radius of convergence. We note that we might want to make specific choices of normalization for the coefficients $c_n$, and that's why we have introduced an additional (in general complex) number $\mathsf{S}$ in (A.5), which is called a *Stokes constant*. We can now associate to the expansion around the singularity (A.5) the following factorially divergent series

$$\varphi_\omega(z) = \sum_{n\geq 1}c_n z^n, \qquad c_n = (n-1)!\tilde{c}_{n-1}, \tag{A.7}$$

which can be regarded as the inverse Borel transform of (A.6). Therefore, given the formal power series $\varphi(z)$, the expansion of its Borel transform around its singularities generates additional formal power series:

$$\varphi(z) \to \{\varphi_\omega(z)\}_{\omega\in\Omega}, \tag{A.8}$$

where $\Omega$ labels the set of singular points. We will call the set of functions $\varphi_\omega(z)$, together with their Stokes constants $\mathsf{S}_\omega$, the *resurgent structure* associated to the original series $\varphi(z)$. Although we have illustrated this construction in the case of logarithmic singularities (A.5), it also holds for other class of singularities, like poles or algebraic branch cuts of the form $\xi^s$.

A basic result of resurgence is that the new series $\varphi_\omega(z)$ "resurge" in the original series through the behavior of the coefficients $a_k$ when $k$ is large. Let $\varphi(z)$ be a resurgent function, and let $A$ be the singularity of the Borel transform which is closest to the origin in the complex $\zeta$ plane (we will assume for simplicity that there is only one, although the generalization is straightforward). Let us assume that the behavior near this singularity is as in (A.5), with $\zeta_\omega = A$. Then, the coefficients $a_k$ have the following asymptotic behavior,

$$a_k \sim \frac{\mathsf{S}}{2\pi}\sum_{n\geq 1}A^{-k+n}c_n\Gamma(k-n), \qquad k \gg 1. \tag{A.9}$$

For the purposes of these lectures, the most convenient way to encode the information in the singularities is through the *Stokes automorphism*. To introduce it we first need some definitions.

Let $\zeta_\omega$ be a singularity of $\widetilde{\varphi}(\zeta)$. A ray in the Borel plane which starts at the origin and passes through $\zeta_\omega$ is called a *Stokes ray*. It is of the form $\mathrm{e}^{\mathrm{i}\theta}\mathbb{R}_+$, where $\theta = \arg(\zeta_\omega)$. Note that a Stokes ray might pass through many singularities. A typical situation is that we have an infinite sequence of singularities on the ray, of the form $\ell\mathcal{A}$ with $\ell \in \mathbb{Z}_{>0}$.

Let $\varphi(z)$ a Gevrey-1 formal power series. If $\widetilde{\varphi}(\zeta)$ analytically continues to an $L^1$-analytic function along the ray $\mathcal{C}^\theta := \mathrm{e}^{\mathrm{i}\theta}\mathbb{R}_+$, we define its *Borel resummation* along the direction $\theta$ by

$$s_\theta(\varphi)(z) = a_0 + \int_{\mathcal{C}^\theta}\widetilde{\varphi}(\zeta)\mathrm{e}^{-\zeta/z}\mathrm{d}\zeta. \tag{A.10}$$

Let us first note that, if $s_\theta(\varphi)(z)$ exists, its asymptotic behavior for small $z$ can be obtained by expanding the integrand and integrating term by term:

$$s_\theta(\varphi)(z) \sim \sum_{n\geq 0}a_n z^n. \tag{A.11}$$

This is the formal power series that we started with. Therefore, if we are lucky, Borel resummation produces an actual function which reproduces the original series. It is then a way to "make sense" of our original formal power series.

If we vary $\theta$ and we do not encounter singularities of $\widehat{\varphi}$, the function $s_\theta(\varphi)(z)$ is locally analytic. However, when $\theta$ is the direction of a Stokes ray, the Borel resummation is not well defined. In fact, as $\theta$ crosses a Stokes ray, it has a discontinuity. To define this discontinuity more precisely, we introduce *lateral Borel resummations*.

Let $\varphi(z)$ be a resurgent function, and let $\mathcal{C}_\pm^\theta$ be contours starting at the origin and going slightly above (respectively, below) the Stokes ray, in such a way that $\mathcal{C}_+^\theta - \mathcal{C}_-^\theta$ is a clockwise contour. Then, the lateral Borel resummations of $\varphi(z)$ are defined as

$$s_{\theta\pm}(\varphi)(z) = \int_{\mathcal{C}_\pm^\theta} \widetilde{\varphi}(\zeta) \mathrm{e}^{-\zeta/z} \mathrm{d}\zeta. \tag{A.12}$$

The discontinuity is then defined by

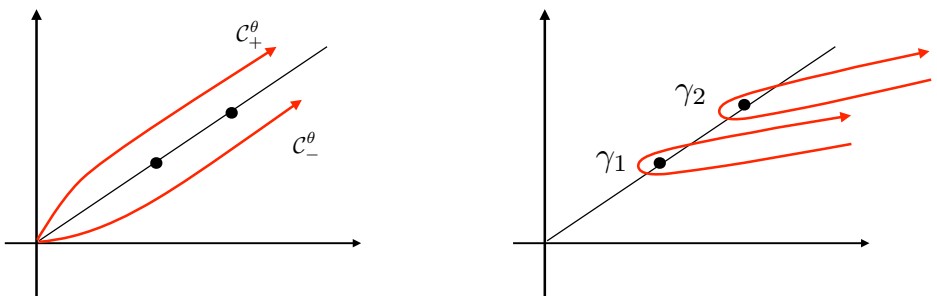

**Figure 10**: Contour deformation in the calculation of the discontinuity.

$$\mathrm{disc}_\theta(\varphi)(z) = s_{\theta+}(\varphi)(z) - s_{-\theta}(\varphi)(z). \tag{A.13}$$

Note that, since $s_{\theta\pm}(\varphi)(z)$ have the same asymptotics for small $z$, given in (A.11), the discontinuity must be invisible in a conventional asymptotic expansion. As we will now see, this difference is *exponentially small* and closely related to the local structure of the Borel transform. Indeed, let us assume e.g. that $\varphi(z)$ is a simple resurgent function, with logarithmic singularities at $\zeta_\omega$ in the Stokes ray, where $\omega \in \Omega$. The difference between the two contours $\mathcal{C}_+^\theta - \mathcal{C}_-^\theta$ can be deformed into a sum of Hankel-like contours $\gamma_\omega$ around the logarithmic branch cuts. We then have, for each $\omega$,

$$\oint_{\gamma_\omega} \widetilde{\varphi}(\zeta) \mathrm{e}^{-\zeta/z} \mathrm{d}\zeta = -\frac{\mathrm{e}^{-\zeta_\omega/z}}{2\pi} \int_{\mathcal{C}_-^\theta} (\log(\xi) - \log(\xi) - 2\pi\mathrm{i}) \, \widetilde{\varphi}_\omega(\xi) \mathrm{e}^{-\xi/z} \mathrm{d}\xi, \tag{A.14}$$

where in the first line we have written $\zeta = \zeta_\omega + \xi$. Therefore

$$s_{\theta+}(\varphi)(z) - s_{\theta-}(\varphi)(z) = \sum_{\omega \in \Omega} \oint_{\gamma_\omega} \widetilde{\varphi}(\zeta) \mathrm{e}^{-\zeta/z} \mathrm{d}\zeta = \mathrm{i} \sum_{\omega \in \Omega} \mathrm{e}^{-\zeta_\omega/z} \int_{\mathcal{C}_-^\theta} \widetilde{\varphi}_\omega(\xi) \mathrm{e}^{-\xi/z} \mathrm{d}\xi$$
$$= \mathrm{i} \sum_{\omega \in \Omega} \mathrm{e}^{-\zeta_\omega/z} s_-(\varphi_\omega)(z). \tag{A.15}$$

This formula holds for more general types of singularities, with appropriate definitions of the trans-series $\varphi_\omega(z)$. An example is given in (3.4).

The expression (A.15) involves (possible infinite) sums of the formal series $\varphi_\omega(z)$ with an exponentially small prefactor $\mathrm{e}^{-\zeta_\omega/z}$. These objects are called *trans-series*. More formally, let $\varphi_\omega(z)$ be resurgent functions. A *trans-series* is a (possibly infinite) formal linear combination of formal power series

$$\Phi(z; \boldsymbol{C}) = \sum_\omega C_\omega \mathrm{e}^{-\zeta_\omega/z} \varphi_\omega(z), \tag{A.16}$$

where $\boldsymbol{C} = (C_{\omega_1}, \cdots)$ is a (possibly infinite) vector of complex numbers.

The result (A.15) involves Borel resummed trans-series, but it is useful to rewrite it as a relation between formal trans-series themselves. If we regard lateral Borel resummations as operators, we introduce the *Stokes automorphism* along the ray $\mathcal{C}^\theta$, $\mathfrak{S}_\theta$, as

$$s_{\theta+} = s_{\theta-}\mathfrak{S}_\theta. \tag{A.17}$$

Then, we can write (A.15) as

$$\mathfrak{S}_\theta(\varphi) = \varphi + \mathrm{i} \sum_{\omega \in \Omega} \mathsf{S}_\omega \mathrm{e}^{-\zeta_\omega/z} \varphi_\omega(z). \tag{A.18}$$

We would like to emphasize that, although we have introduced the Stokes automorphism by using lateral Borel resummations, the expression (A.18) just collects the local information of the Borel transform near the Borel singularities on a ray. In fact, in Écalle's theory, the Stokes automorphism can be defined in terms of the so-called alien derivatives [148], which encode this local information and do not involve resummations.

We will now state a principle of *semiclassical decoding*.

**Definition A.2.** (Semiclassical decoding). Let $f(z)$ be a function with the asymptotic expansion

$$f(z) \sim \varphi(z) = \sum_{n \geq 0} a_n z^n. \tag{A.19}$$

We say that $f(z)$ admits a *semiclassical decoding* if $\varphi(z)$ can be promoted to a trans-series $\Phi(z; \boldsymbol{C})$, which is lateral Borel summable, and such that

$$f(z) = s_\pm(\Phi)(z; \boldsymbol{C}_\pm) \tag{A.20}$$

for some vectors of complex constants $\boldsymbol{C}_\pm$.

When semiclassical decoding holds, one recovers the exact information by just considering Borel-resummed trans-series. Conversely, we can think about resummed trans-series as building blocks of non-perturbative answers.

The simplest situation corresponds to the case in which $C = 0$, there are no singularities along the positive real axis, and the Borel resummation of the perturbative series reproduces the exact result. This is famously the case for the perturbative series of the quartic oscillator, as we mentioned in section 1.

## B Faddeev's quantum dilogarithm

Faddeev's quantum dilogarithm $\Phi_{\mathsf{b}}(x)$ is defined by [58]

$$\Phi_{\mathsf{b}}(x) = \frac{(\mathrm{e}^{2\pi\mathsf{b}(x+c_{\mathsf{b}})}; q)_\infty}{(\mathrm{e}^{2\pi\mathsf{b}^{-1}(x-c_{\mathsf{b}})}; \tilde{q})_\infty}, \tag{B.1}$$

where

$$q = \mathrm{e}^{2\pi \mathrm{i} \mathsf{b}^2}, \qquad \tilde{q} = \mathrm{e}^{-2\pi \mathrm{i} \mathsf{b}^{-2}}, \qquad \mathrm{Im}(\mathsf{b}^2) > 0 \tag{B.2}$$

and

$$c_{\mathsf{b}} = \frac{\mathrm{i}}{2}\left(\mathsf{b} + \mathsf{b}^{-1}\right). \tag{B.3}$$

Explicitly, this gives

$$\Phi_{\mathsf{b}}(x) = \prod_{n=0}^{\infty} \frac{1 - \mathrm{e}^{2\pi \mathsf{b}(x+c_{\mathsf{b}})} q^n}{1 - \mathrm{e}^{2\pi \mathsf{b}^{-1}(x-c_{\mathsf{b}})} \tilde{q}^n}. \tag{B.4}$$

From this infinite product representation one deduces that $\Phi(x)$ is a meromorphic function of $x$ with

$$\text{poles: } c_{\mathsf{b}} + \mathrm{i}\mathbb{N}\mathsf{b} + \mathrm{i}\mathbb{N}\mathsf{b}^{-1}, \qquad \text{zeros: } -c_{\mathsf{b}} - \mathrm{i}\mathbb{N}\mathsf{b} - \mathrm{i}\mathbb{N}\mathsf{b}^{-1}. \tag{B.5}$$

An integral representation in the strip $|\mathrm{Im}z| < |\mathrm{Im}\, c_{\mathsf{b}}|$ is given by

$$\Phi_{\mathsf{b}}(x) = \exp\left(\int_{\mathbb{R}+\mathrm{i}\epsilon} \frac{\mathrm{e}^{-2\mathrm{i}xz}}{4\sinh(z\mathsf{b})\sinh(z\mathsf{b}^{-1})} \frac{\mathrm{d}z}{z}\right). \tag{B.6}$$

Remarkably, this function admits an extension to all values of $\mathsf{b}$ with $\mathsf{b}^2 \notin \mathbb{R}_{\leq 0}$. A useful property is

$$\Phi_{\mathsf{b}}(x)\,\Phi_{\mathsf{b}}(-x) = \mathrm{e}^{\pi \mathrm{i} x^2}\,\Phi_{\mathsf{b}}(0)^2, \qquad \Phi_{\mathsf{b}}(0) = \left(\frac{q}{\tilde{q}}\right)^{\frac{1}{48}} = \mathrm{e}^{\pi \mathrm{i}\left(\mathsf{b}^2 + \mathsf{b}^{-2}\right)/24}. \tag{B.7}$$

In addition, when $\mathsf{b}$ is either real or on the unit circle, we have the unitarity relation

$$\overline{\Phi_{\mathsf{b}}(x)} = \frac{1}{\Phi_{\mathsf{b}}\left(\overline{x}\right)}. \tag{B.8}$$

From the product representation (B.4) it follows immediately that Faddeev's quantum dilogarithm is a quasi-periodic function. Explicitly, it satisfies the equations

$$\frac{\Phi_{\mathsf{b}}(x + c_{\mathsf{b}} + \mathrm{i}\mathsf{b})}{\Phi(x + c_{\mathsf{b}})} = \frac{1}{1 - q\mathrm{e}^{2\pi \mathsf{b}x}} \tag{B.9a}$$

$$\frac{\Phi_{\mathsf{b}}(x + c_{\mathsf{b}} + \mathrm{i}\mathsf{b}^{-1})}{\Phi_{\mathsf{b}}(x + c_{\mathsf{b}})} = \frac{1}{1 - \tilde{q}^{-1}\mathrm{e}^{2\pi \mathsf{b}^{-1}x}}. \tag{B.9b}$$

When $\mathsf{b}$ is small, we can use the folllowing asymptotic expansion,

$$\log \Phi_{\mathsf{b}}\left(\frac{x}{2\pi \mathsf{b}}\right) \sim \sum_{k=0}^{\infty} \left(2\pi \mathrm{i}\mathsf{b}^2\right)^{2k-1} \frac{B_{2k}(1/2)}{(2k)!} \mathrm{Li}_{2-2k}(-\mathrm{e}^x), \tag{B.10}$$

where $B_{2k}(z)$ is the Bernoulli polynomial.

When

$$\mathsf{b}^2 = \frac{M}{N} \tag{B.11}$$

is a rational number, Faddeev's quantum dilogarithm can be written in terms of the conventional dilogarithm [151]. In particular, when $M = N = 1$, one finds

$$\Phi_1(x) = \exp\left[\frac{\mathrm{i}}{2\pi}\left(\mathrm{Li}_2\left(\mathrm{e}^{2\pi x}\right) + 2\pi x \log\left(1 - \mathrm{e}^{2\pi x}\right)\right)\right]. \tag{B.12}$$

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
