# Peer review of "Les Houches lectures on non-perturbative topological strings"

_SciPost Physics Lecture Notes_

## Round 2 · Referee Report · Anonymous (Referee 1) · 2025-4-11

Report

These lecture notes offer an excellent and timely overview of the current status and modern developments in non-perturbative topological string theory, providing both a pedagogical introduction and a comprehensive perspective on recent advances in the field.

Requested changes

I have only one minor comment concerning Remark 2.4. To the best of my knowledge, the proof in [44] applies only for \epsilon_1, \epsilon_2 > 0. The convergence in the topological string phase was established prior to [44] in [arXiv:1608.02566], Proposition 3.1.

Recommendation

Publish (surpasses expectations and criteria for this Journal; among top 10%)

  • validity: top
  • significance: top
  • originality: -
  • clarity: top
  • formatting: excellent
  • grammar: excellent

Author:  Marcos Mariño  on 2025-06-26  [id 5601]

(in reply to Report 1 on 2025-04-11)

Thank you very much for your positive feedback. You are right about ref. [44], I will definitely add the relevant reference that you point out in the revised version.

---

## Round 2 · Referee Report · Anonymous (Referee 2) · 2025-6-20

Report

The lectures under review offer a very useful review of some important developments in topological string theory of the last decade.

The presentation is often terse, but usually clearly written. I'd expect it to be useful for young researchers or PhD students looking for a fast track into this field of research.

I'd certainly like to recommend these lecture notes for publication.

Requested changes

  • P.2, bottom, "...and a non-rigorous ... is typical in QFT". Confusing or misleading formulation

  • P.12, "...is obtained as a global function on the moduli space...": This is very confusing, as just a little later the dependence of the pre-potential on choices such as a symplectic basis is observed, playing a very important role later on. On page 29 it is written that "The free energies in a given frame F_g are not globally defined on the moduli space, and they are analytic only on a region...". While it may well be true that local coordinates defining a particular frame may admit an analytic continuation to a much larger domain, it seems better to stress the frame dependence rather than the "global" nature.

P. 23, "...that t is "quantised" in units of g_s.", and similar bottom of P.32: I can't help observing a tension with the fact that later it plays an important role to have functions that are entire as function of the moduli, and not just defined on a lattice. To my mind it remains obscure what the precise meaning of the "quantisation" of t in this context is supposed to be. Having finite shifts alone can hardly be enough to justify such far-reaching proposals. A continuation away from the integer multiples of g_s does exist, and it plays an important role in the story.

P. 34, "In this correspondence, the dual of the string theory is not a field theory, but a one-dimensional quantum-mechanical model." -- A very confusing statement. The duality discussed here is of very different nature than other dualities in QFT and string theory such as S-duality of N=4 SYM, or AdS-CFT. Such dualities claim dual descriptions of a QFT or string theory as a whole in terms of either another QFT, or another string theory. One certainly does not expect the ST to capture all of the string theory compactified on CY.

P. 35, "different canonical forms of the curve": Shouldn't the form be unique if it is canonical? And functional-difference -> finite difference

Recommendation

Publish (surpasses expectations and criteria for this Journal; among top 10%)

  • validity: top
  • significance: top
  • originality: -
  • clarity: top
  • formatting: excellent
  • grammar: good

Author:  Marcos Mariño  on 2025-06-26  [id 5602]

(in reply to Report 2 on 2025-06-20)

Thank you very much for your feedback and the detailed reading. Let me make some comments on your observations.

1) Concerning your observation on my statements in p.2: what I mean when I say "...and a non-rigorous ... is typical in QFT" is explained in the paragraph immediately after this sentence. Would it be possible to have a more explicit explanation of what is confusing or misleading in this paragraph?

2) I agree that this requires clarification. What I mean here is that in the B model one can write expressions for the genus g free energies in terms of propagators and global functions which are really global and universal, frame-independent. Expressions in different frames are then obtained by different choices of the form of the propagator.

3) I also agree that this requires further clarification. One important comment that could be clarifying is that the expressions obtained for the multi-instantons in the topological string are very similar to the multi-instantons in matrix models obtained by eigenvalue tunneling. The latter involve an actual quantization since one has to "tunnel" integer numbers of eigenvalues. The matrix model realization of the free energies explained later in the lectures also leads to a realization of this "quantization". I will try to clarify these issues and the existing tension in the revised version.

4) I'm not sure why one should be so restrictive concerning the use of the word "duality". In addition, there is evidence that in practice the full topological string in the toric case can be described by this quantum mechanical model. The observables of topological string are closed string amplitudes F_g and open string amplitudes involving toric D-branes. For the first one, there is clearly a complete description in terms of spectral traces. For the second one, there is also evidence that they can be described by wavefunctions in the quantum mechanics. Perhaps where I concur with the referee is that this duality can not be explained right now by a more fundamental string theory construction, for example. This could be added as a qualification.

5) For this last comment, I refer in the lectures to the paper [15] for explaining what is meant by "canonical" forms, but I agree that the phrasing is confusing. I can definitely change functional-> finite.

---

## Editorial Decision

accepted_in_target_journal